# Label-free human skin imaging with enhanced molecular contrast via time-resolved fluorescence and advanced phasor analysis

Suman Ranjit [1,2,4] ✉, Belen Torrado[1,4], Alexander Vallmitjana[1], Amanda Fedyk Durkin[1], Alexander Dvornikov[1], Anand Ganesan[1,3], Kristen M. Kelly[1,3] & Mihaela Balu [1,3] ✉

Current state-of-the-art clinical skin imaging using label-free multiphoton microscopy (MPM) faces challenges due to limited molecular specificity, which hampers the accurate characterization of skin tissues because of overlapping fluorescence signals from multiple molecular components. In this study, we present a novel approach to enhance molecular contrast in MPM clinical skin imaging by leveraging advanced strategies to effectively unmix the various endogenous fluorophores present in the skin with the performance capabilities of a recently developed imaging platform for in vivo time-resolved fluorescence imaging of human skin. By identifying phasor positions of key endogenous skin fluorophores – such as keratin, melanin, free NADH, and protein-bound NADH – we effectively perform multicomponent unmixing in different skin types and conditions, including those with varying levels of pigmentation and metabolic states. The phasor analysis allows for the mapping and quantification of individual fluorescence species and provides comprehensive insight into the dynamic changes in molecular components associated with distinct clinical conditions. This study highlights the effective use of advanced imaging and phasor analysis to improve label-free molecular contrast in clinical skin imaging, leading to advancements in future research focused on precise and timely assessments of skin conditions and monitoring of therapeutic effects.

Current label-free dermatologic imaging modalities, such as reflectance confocal microscopy (RCM) and optical coherence tomography (OCT), provide high-resolution structural information but lack molecular specificity. RCM can be combined with machine learning[1] to improve diagnostic performance, and fluorescence confocal approaches often require exogenous dyes[2]. These limitations motivate the development of imaging techniques that combine high resolution, molecular contrast, and label-free detection, such as multiphoton microscopy (MPM).

MPM is a widely used imaging technique in research laboratories, valued for its ability to generate high-resolution, three-dimensional maps of biological tissues, making it essential for biological research. Studies combining MPM with fluorescent reporter mouse models and in vivo labeling techniques have provided critical insights into research fields such as immunology[3], neuroscience[4] and cancer biology[5]. However, the inherent limitations of animal models in studying human tissue at the cellular level, particularly skin, have driven efforts to develop MPM as non-invasive imaging tool for clinical diagnostics and treatment monitoring, especially in dermatology[6–10]. Clinical applications of MPM rely on label-free endogenous fluorescence and harmonic generation signals, suitable for in vivo, non-invasive imaging. Despite this advantage, label-free imaging is often limited by insufficient molecular specificity, posing challenges in distinguishing various cell populations within tissues.

Time-resolved fluorescence detection has been explored to enhance molecular contrast[11], showing promise particularly in the selective detection of melanin, exploiting melanin's relatively short fluorescence lifetime compared to other endogenous skin fluorophores. This method has demonstrated utility for in vivo melanin quantification in human skin[12–14] and holds potential for non-invasive diagnosis of melanoma[15].

[1]Beckman Laser Institute and Medical Clinic, University of California Irvine, Irvine, CA, USA. [2]Department of Oncology, Georgetown University Medical Center, Washington, DC, USA. [3]Department of Dermatology, University of California Irvine, Irvine, CA, USA. [4]These authors contributed equally: Suman Ranjit, Belen Torrado. ✉e-mail: suman.ranjit@georgetown.edu; mbalu@uci.edu

Despite these advancements, several limitations of previously reported studies hinder the efficiency of clinical fluorescence lifetime imaging (FLIM) of the skin and restrict its scalability for broader validation and use. These limitations include slow scanning speeds necessary for accurate fluorescence lifetime detection, which impedes the rapid imaging essential for clinical workflows. Additionally, the limited field of view of current clinical FLIM systems hinders the ability to capture adequate imaging samples of lesions, which is essential for accurate assessment in dermatological evaluations. Furthermore, robust methodologies to decompose and analyze multiple molecular signals observed in vivo in a label-free context are lacking, complicating accurate tissue characterization.

In human skin, several molecular components contribute to the autofluorescence signals detected by label-free MPM in vivo imaging, including Nicotinamide Adenine Dinucleotide Hydride (NADH), Flavin Adenine Dinucleotide (FAD), keratin and melanin in the epidermis, as well as collagen and elastin in the dermis. Most of these molecular components have overlapping fluorescence spectra and fluorescence lifetime signatures, making signal analysis and understanding the contributions of individual molecular components extremely complex.

In this study, we demonstrate significant enhancement in label-free molecular contrast for in vivo MPM skin imaging in a clinical setting by applying time-resolved fluorescence detection in conjunction with advanced phasor analysis. This analysis is integrated into an open-source software platform, GSLab, recently developed in our lab[16]. This approach enables us to analyze complex skin autofluorescence signatures and effectively perform multicomponent unmixing of key endogenous skin fluorophores, including keratin, melanin, free and protein-bound NADH. These recent advancements in multicomponent unmixing, involving higher harmonic phasor calculations[17–21], allow us to map individual fluorescence species across images. Fractional intensity distributions can be computed by attributing photon contributions from each species to the total fluorescence at each pixel[22]. This capability enables the assessment of changes in specific molecular components associated with different clinical conditions, providing valuable insights into skin health, disease states and metabolic changes.

We apply this analytical approach for the first time to images acquired in vivo, in a clinical setting by using the fast, large-area multiphoton exoscope (FLAME), an advanced imaging platform developed by our group with specifications optimized for clinical skin imaging (Supplementary Fig. 1)[8]. FLAME enables rapid acquisition of high-resolution depth-resolved images over several squared millimeter area while maintaining sub-cellular resolution (0.5–1 μm) within less than a minute. This combination of effective imaging capabilities and robust molecular analysis positions FLAME as a powerful tool for clinical skin imaging, promising significant advancements in non-invasive dermatological diagnostics and the monitoring of therapeutic effects.

This manuscript focuses on identifying the phasor positions of skin fluorophores in epidermis, a task that is strategically informed by the unique biological characteristics associated with specific skin types and conditions. Additionally, we assess the reliability of the multicomponent unmixing process by analyzing variations in these phasor positions at specific skin depths, across various skin types, and under different conditions characterized by varying levels of pigmentation and metabolic changes. Ultimately, we emphasize the significance of the unmixing process in a clinically relevant scenario, highlighting its potential impact on the understanding and diagnosis of skin-related conditions.

## Results
The autofluorescence signal captured in patient skin during in vivo imaging is significantly more complex than traditional cell autofluorescence measurements conducted in cell culture. In addition to the contributions from free and protein-bound NADH and FAD, other autofluorescence components excited by 785 nm wavelength of FLAME include keratin, melanin, elastin and collagen[23]. In the epidermis, keratin and melanin are prominent fluorescent species[24,25], with melanin being widely distributed throughout

the epidermis, particularly in individuals with darker skin types[14,26–28]. The following results leverage the capabilities of the FLAME platform for in vivo time-resolved fluorescence imaging of human skin, employing advanced strategies to unmix the various endogenous fluorophores present in the epidermis. The molecular components and their corresponding fluorescence in the dermis add another layer of complexity, which will be discussed in a separate publication.

### Depth resolved phasor FLIM analysis of human skin based on in vivo FLAME imaging
We employed the FLAME imaging platform to distinguish key molecular components in human skin based on fluorescence lifetime and skin layer location. Fig. 1 highlights FLAME's capability to provide depth-resolved images of human skin, acquired both in vivo and ex vivo, alongside the analysis approach developed to characterize their fluorescence lifetime properties. These images are presented as vertical optical sections (acquired ex vivo) or en-face optical sections (acquired in vivo), capturing skin layers, including the stratum corneum, epidermis, basal layer, dermal-epidermal junction (DEJ), and the dermis (Fig. 1a).

Fluorescence lifetime detection was performed to analyze fluorescence lifetime properties of skin endogenous fluorophores across two spectral channels: A0 (405 - 506 nm) and A1 (506 - 610 nm). Two-photon excited fluorescence (TPEF) decays were transformed into phasor plots used to quantify fluorescence lifetimes and enhance molecular contrast by color-coding based on the measured phase lifetimes (Fig. 1b).

Reference phasor positions for the major skin fluorophores were established by combining published data with our own measurements (Fig. 1c). Free NADH and free FAD were measured in solution with our system and validated against literature values[29–32], while protein-bound NADH and FAD lifetime were adopted from the literature[30–32]. In contrast, keratin and melanin lifetimes were determined in vivo, where specific physiological conditions allowed each to dominate the autofluorescence signal: keratin in the stratum corneum of a vitiligo subject (melanin absent) and melanin in basal keratinocytes of a highly pigmented skin type V subject. This strategy enabled us to obtain intrinsic phasor signatures of these fluorophores directly in human skin tissue, rather than relying solely on solution-based values. The two detection spectral windows, A0 and A1, were selected based on the TPEF emission spectra of these molecular species (Fig. 1d)[33]. For the unmixing analysis, however, we restricted detection to the A0 channel (405–506 nm), where FAD contribution is minimal, in order to reduce analysis complexity. Thus, while FAD was characterized for reference, it was not included in the unmixing analysis. Instead, we focused on separating four distinct components: free and protein-bound NADH from keratin and melanin (Fig. 1e). We generated fractional intensity maps that provide spatial resolution of photon distributions associated with key cellular features and quantified their contributions. This approach leverages physiologically relevant in vivo lifetime signatures to achieve label-free molecular contrast in human skin, as described in the following sections.

### Melanin contribution to phasor plot distribution across human skin epidermis
To understand how melanin impacts the phasor plot distribution in different layers of the epidermis, we imaged in vivo the skin of three volunteers with varying levels of melanin: high melanin content (skin type V), moderate melanin content (skin type III), and minimal melanin content (vitiligo, a chronic autoimmune disorder characterized by depigmented skin patches[34]). Melanin exhibits a red-shifted fluorescence spectrum compared to NADH and keratin, with an emission maximum around 550 nm (Fig. 1d). Under our detection settings, most of the melanin fluorescence is captured in the A1 channel (506–610 nm).

To explore the phasor position of melanin, we imaged the forearm of a volunteer with skin type V, where melanin, particularly in the basal layer, is the primary contributor to the fluorescence signal. Fig. 2a presents representative images acquired at varying depths in the epidermis of skin type V, along with their corresponding phasor distributions. These images highlight

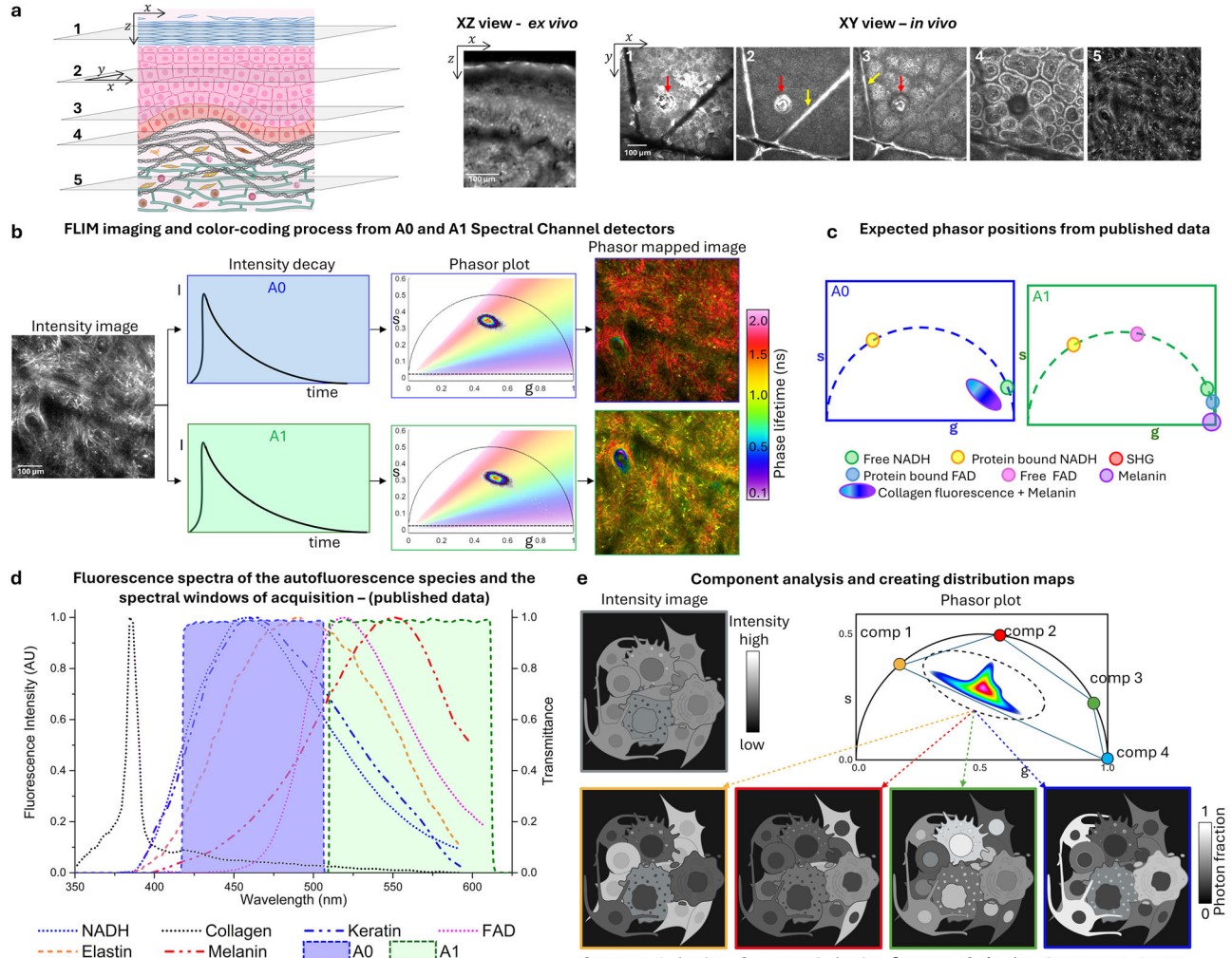

**Fig. 1 | Multiphoton Microscopy of Skin Layers and Phasor-Based FLIM Analysis. a** Schematic representation of skin layers, including the stratum corneum (1), epidermis (2), basal layer (3), dermal-epidermal junction (DEJ, 4) and dermis (5); MPM images were acquired ex vivo (XZ optical section) and in vivo (en-face optical sections), highlighting: (1) uniform keratin distribution in the stratum corneum, (2) skin folds (yellow arrow), a hair follicle (red arrow) and keratinocytes in the epidermis, (3) bright pigmented keratinocytes above the dermal papillae tips in the basal layer, (4) pigmented keratinocytes surrounding dermal papillae at the DEJ, and (5) collagen and elastin fibers in the dermis. Scale bar is 100 μm. **b** Time-resolved MPM image of the dermis (in vivo, acquired using FLAME on a volunteer's forearm), fluorescence decay signals detected on two detection channels A0 (405 –

506 nm) and A1 (506-610 nm) along with corresponding phasor plots with color coding based on the phase angle. **c** Fluorescence lifetime phasor positions of major molecular skin components. **d** Endogenous TPEF and SHG emission spectra of skin components, overlaid with FLAME's detection spectral windows (recreated from[33]). **e** Schematic depiction of phasor analysis highlighting the unmixing of four distinct tissue components based on their phasor signatures. Fractional intensity maps are generated for each component, visualizing their spatial distribution. Highlighted regions indicate representative areas where each component predominate, aiding interpretation of the unmixing results. Graphics in (**a**) and (**e**) were created with biorender.com (Created in BioRender. Ranjit, S. (2025) https://BioRender.com/thvhn58) and Powerpoint.

the gradual increase in melanin contribution from the stratum corneum to the basal layer. In the stratum corneum, fluorophores with longer fluorescence lifetimes play a significant role, while in the basal layer, melanin becomes the major contributor, characterized by shorter fluorescence lifetimes[27]. In skin type V, the melanin-rich areas containing basal keratinocytes and melanocytes show a phasor signature including a very short fluorescence lifetime contribution, close to zero nanoseconds on the phasor plot. This phasor position identifies the melanin component in skin as observed through FLAME imaging (Fig. 2a, column 5). The calculated phasor coordinates are presented in Supplemental Table 1.

Representative images from the three volunteers are shown in Fig. 2a (column 2 and 4) for skin type V, Fig. 2b for skin type III and Fig. 2c for vitiligo. Images were captured at two epidermal depths: the top layer (stratum corneum or stratum granulosum) and the basal layer (stratum basale). For each image, we examined the corresponding phasor plot distribution to evaluate shifts in fluorescence lifetimes of endogenous

fluorophores relative to the expected lifetime value of melanin (blue circle), marked on each plot.

Across the epidermis, the phasor distributions showed a shift from longer fluorescence lifetimes associated with non-melanin skin fluorophores to shorter lifetimes associated with melanin, the predominant component in melanocytes and keratinocytes of basal layer. At both depths, the extent of the shift depended on melanin levels: in skin type V, the phasor distribution was concentrated toward shorter lifetime values; in skin type III, it was balanced between short and long lifetimes; and in vitiligo, it was shifted more toward longer lifetimes. As expected, the melanin contribution was significantly higher in the basal layer compared to the stratum corneum. To better illustrate melanin's presence and contribution, we provide phasor-mapped images, generated by multiplying the intensity images with the phasor color map.

We also assessed melanin's contribution to the fluorescence detected in the A0 channel (405–506 nm) at the two selected epidermal depths. Phasor

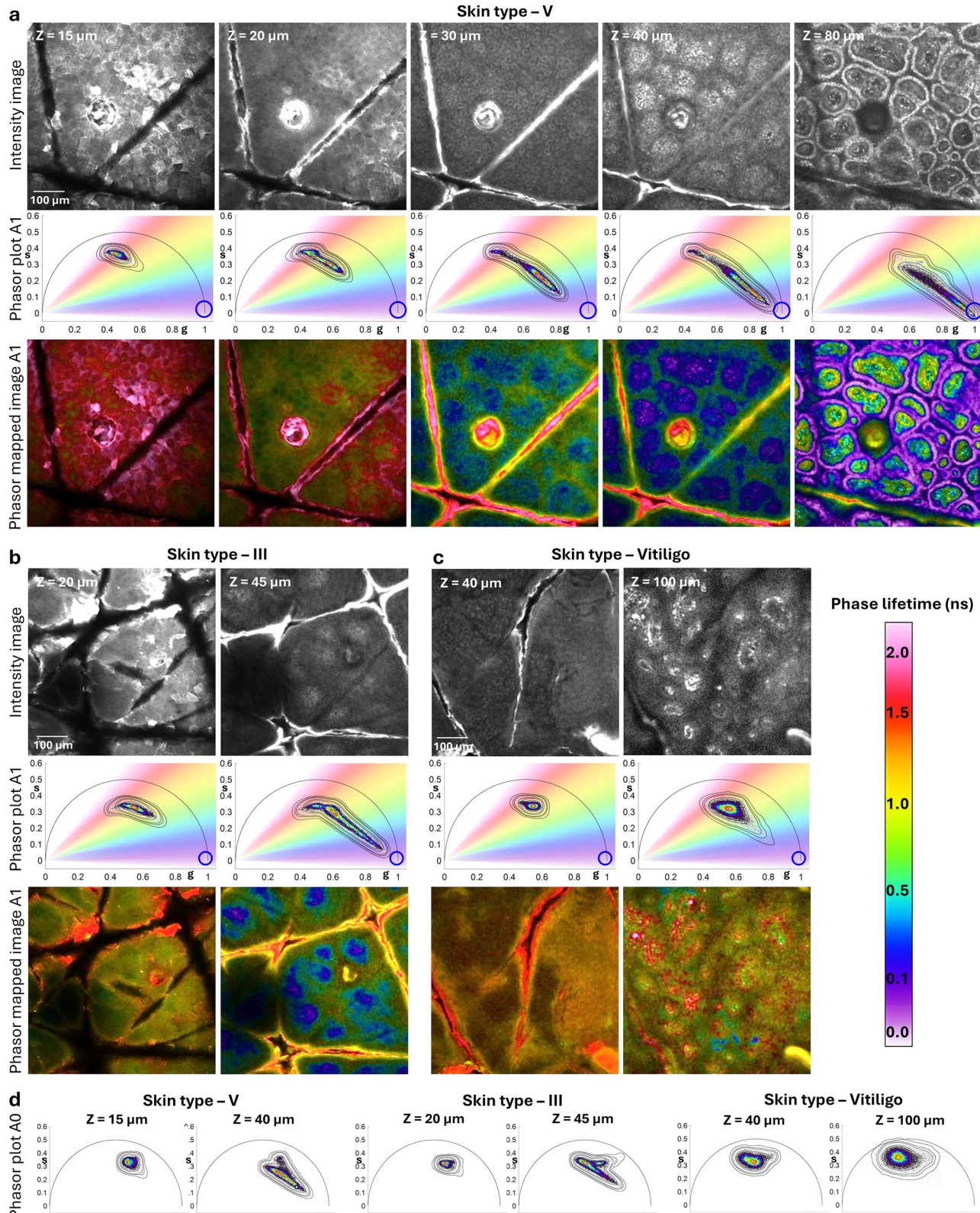

**Fig. 2 | Depth-dependent melanin contribution to phasor signatures across different skin types.** Representative data from three volunteers with varying melanin levels: **skin type V** (**a**), **skin type III** (**b**) and **Vitiligo** (**c**). For each skin type, rows display: (1) intensity images, (2) phasor plots from A1 detection channel (506-610 nm) images highlighting shifts in fluorescence lifetime, (3) color-coded phasor-mapped images from the A1 channel showing phase lifetime variations. **a Skin type V:** Depth-resolved intensity images and phasor shifts, showing a gradual transition toward melanin's expected fluorescence lifetime (blue circle) at deeper skin layers.

Directional changes in phasor positions are visible from the surface to the DEJ. Color-coded phasor-mapped images highlight increasing melanin contribution with depth, represented by blue and purple tones. **b, c Skin type III and vitiligo:** Images captured at two representative epidermal depths: upper layers (stratum corneum or stratum granulosum) and basal layer for comparative analysis. **d A0 channel analysis:** Phasor plots from the A0 detection channel (405–506 nm) highlighting shifts in fluorescence lifetime at the previously selected epidermal depths. Scale bars are 100 μm.

distributions in the A0 detection channel (Fig. 2d) exhibited similar behavior to those in the A1 channel but did not shift as strongly toward short lifetimes. This can be attributed to the reduced contribution of melanin in the A0 channel, which primarily captures signals from other fluorophores.

## Keratin contribution to phasor plot distribution across human skin epidermis

Previous studies have shown that the keratin fluorescence peaks at ~465 nm (Fig. 1d)[24,33]. Under our detection settings, most keratin fluorescence is captured in the A0 channel (405–506 nm).

To assess keratin's contribution to the phasor plot distribution in the epidermis, we analyzed depth-resolved images from three volunteers with different skin types. Representative in vivo images from the three volunteers are shown in Fig. 3a for vitiligo, Fig. 3b for skin type V and Fig. 3c (columns 1 and 4) for skin type III. Images were captured at two epidermal depths: the top layer (stratum corneum or stratum granulosum) and the basal layer (stratum basale). For each image, we examined the corresponding phasor plot distribution to evaluate shifts in the fluorescence lifetimes of endogenous fluorophores relative to the expected lifetime value of keratin, marked on each plot.

To explore the keratin phasor position, we focused on the vitiligo lesion, where the absence of melanin in the stratum corneum allows keratin to be the primary contributor to fluorescence enabling more accurate identification. The phasor distribution derived from the stratum corneum image of the vitiligo lesion reflects a phasor position primarily associated with keratin, as observed during FLAME imaging (Fig. 3a). Notably, this phasor distribution has a comet-like structure with the tail pointing towards a fluorescence lifetime component of approximately 1.1 ns, which aligns closely with previously reported fluorescence lifetimes of keratin contribution in keratinocytes[35]. While we will consider this value as the fluorescence lifetime for keratin in our subsequent analysis, it is important to note that this phasor distribution suggests that keratin exhibits a multi-exponential decay characteristic. This complexity is likely attributed to the decay kinetics of keratin, influenced by cross-links arising from keratin glycation[36]. In contrast, the phasor distributions for pigmented normal skin (skin types III and V) the tail of the comet shifts toward shorter fluorescence lifetimes compared to the phasor position observed in vitiligo. This shift is attributed to melanin contributions, which are evident in both the stratum corneum and deeper layers, including the basal layer (Fig. 3b-c).

## Unmixing keratin and melanin and their quantification in epidermal layers

The intrinsic phasor positions of melanin and keratin established from our in vivo measurements, together with fluorescence lifetimes for free and protein-bound NADH obtained from our own measurements and published data, provided the basis for the 4-component unmixing analysis (Table 1). For our analysis, we performed unmixing using fluorescence detected in the A0 channel (405-506 nm) to minimize the FAD contribution to the fluorescence signal.

To evaluate the reliability of the unmixing process, we analyzed changes in melanin and keratin components derived from phasor analysis of in vivo images acquired at various depths of the epidermis in subjects with skin types V (Fig. 4a), III (Fig. 4b), and vitiligo (Fig. 4c).

Fig. 4a-c presents representative depth-resolved images acquired from these subjects, along with maps generated through multicomponent unmixing[17–21]. These maps illustrate the distribution of keratin and melanin throughout the images. Keratin is present throughout the stratum corneum and begins to concentrate in skin folds and pores as imaging progresses deeper into the epidermis. In contrast, melanin is less prominent in the stratum corneum but accumulates significantly in the basal keratinocytes of darker skin types, such as III and V.

We used photon-weighted violin plots of the unmixed component fractions to capture variations in the photon contribution of different fluorescence species across all pixels in images at different skin depths (Fig. 4d, Supplementary Data 1). In these violin plots, the y-axis represents the unmixed photon fraction, defined as the ratio of fluorescence from a fluorescent component in a pixel over the total fluorescence in the pixel. Essentially, fractional intensity reflects the fraction of photons contributed by a particular fluorescent component to the total fluorescence photons at each pixel. We weighted the fractional intensity values by the number of photons collected at each pixel to minimize the influence of pixels with low signal-to-noise ratio (SNR), thereby emphasizing regions with more reliable fluorescence measurements.

As shown in the violin plots in Fig. 4 (Supplementary Data 1), the keratin distribution shifts to lower fraction values with increasing depth in both skin types and vitiligo, indicating a decrease in keratin amount as the depth within the epidermis increases from the superficial layers to the basal layer. In contrast, the melanin distribution increases with depth in skin types III and V. In vitiligo, the melanin distributions maintain their median value close to zero at all depths, reflecting the limited amount of melanin in this skin lesion.

The broadly dispersed violin plots have different implications depending on whether they correspond to low or high fractional intensity values. For instance, the broad melanin distribution in skin type V towards high fractional intensity values, particularly at the basal layer, corresponds to a substantial presence of melanin across most pixels in the image. In contrast, the broad melanin distribution in vitiligo at low fractional intensity values results from a weak melanin fluorescence signal leading to a low SNR. This low SNR contributes to the broad phasor distribution and can produce the negative fractional intensity values in the melanin distributions observed in vitiligo. These negative values arise from mathematical computations and do not hold biological significance, but rather underscore the low SNR.

Some distributions may appear bimodal, featuring two maxima, one at lower fractional intensity values and another at higher values, reflecting low and high contributions of specific fluorescence species in different regions of the image. An example of this is the bimodal distribution of keratin in skin type V, which emerges as depth increases within the epidermis (Fig. 4d). This pattern highlights the significant presence of keratin in the skin folds and hair captured in the image, in contrast to the lower keratin levels found in keratinocytes. The violin plot density reflects keratin distribution heterogeneity, with narrow plot sections corresponding to fewer pixel representing skin folds and hair, while broad plot sections indicate greater pixel populations from keratinocytes. A similar bimodality can be observed for melanin, where the skin folds and hair follicles have lower amount of melanin compared to the melanin-rich keratinocytes in the deeper epidermis.

Keratin and melanin distribution can be effectively mapped over larger skin areas using FLAME and the multicomponent unmixing phasor analysis (Supplementary Fig. 2).

## Applying the unmixing of keratin and melanin in a clinically relevant context

To demonstrate the effectiveness of separating melanin and keratin components in skin for distinguishing critical features in skin lesions, we present representative images (Fig. 5) acquired using FLAME in time-resolved detection mode from a freshly excised shaved biopsy of a lesion diagnosed as macular seborrheic keratosis (SK) following imaging (Fig. 5a). SK typically appears as tan, brown or black, keratotic lesions. Although benign, its appearance can sometimes mimic that of melanoma, especially in atypical cases, making histological examination through biopsy beneficial.

One key histological signature of SK is the presence of "horn cysts", which are keratin-filled pseudocysts (Fig. 5b). Being able to identify this feature through imaging is helpful for distinguishing these lesions from melanoma. Fig. 5c shows a representative image acquired over a large, millimeter-scale area from this lesion, at the dermal-epidermal junction, identifying horn cysts (red arrows). Fig. 5e shows phasor-colored image highlighting different phase lifetime values in different skin structures. The 4-component unmixing phasor analysis generates fractional intensity maps that illustrate the distribution of melanin (blue) and keratin (red)

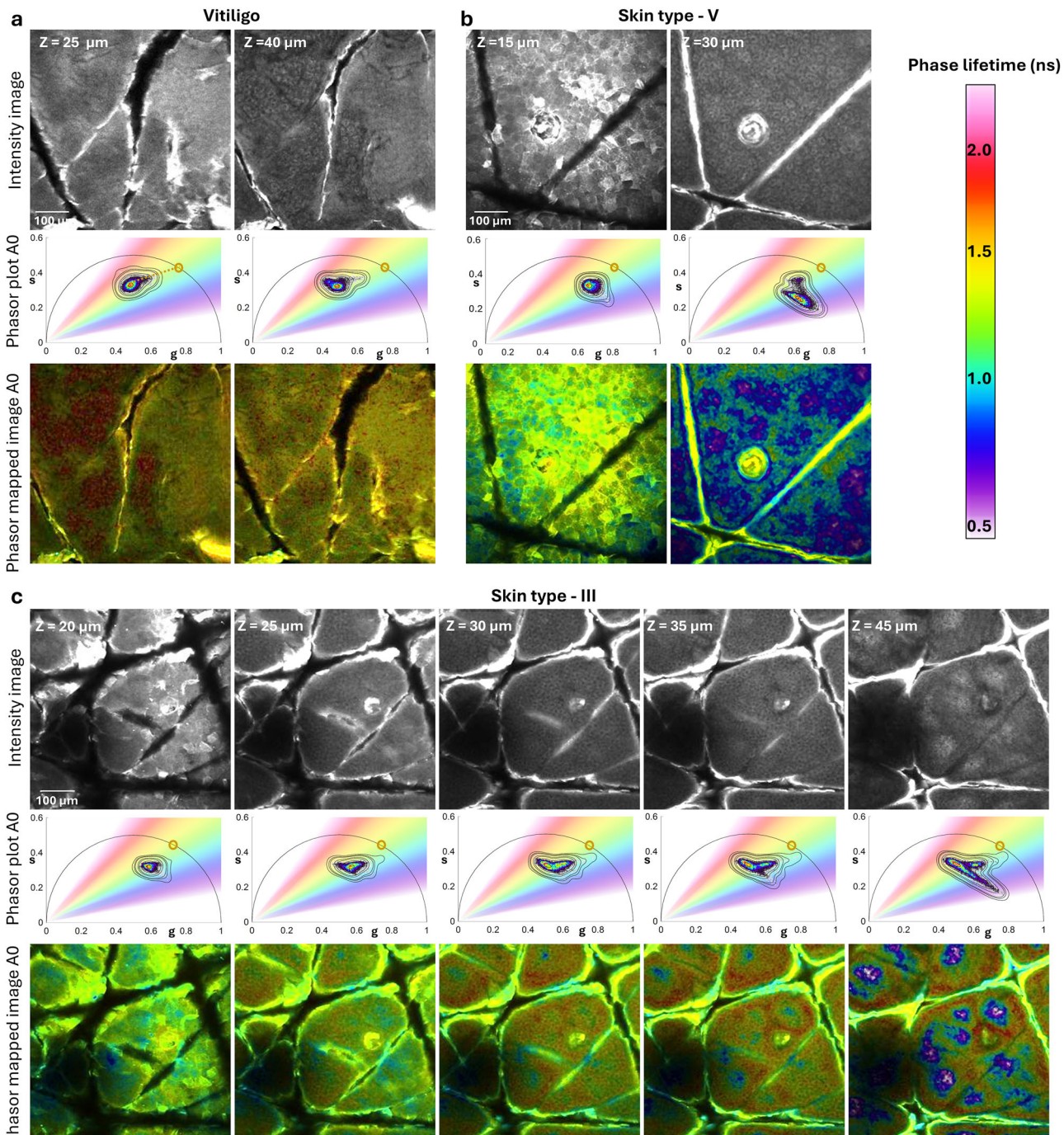

**Fig. 3 | Depth-dependent Keratin contribution to Phasor signatures in skin Types III, V, and Vitiligo.** Representative data from three volunteers with varying melanin levels: vitiligo (**a**), skin type III (**b**), and skin type V (**c**), acquired in <u>A0 channel</u>. For each skin type, row display: (1) intensity images from stratum corneum and stratum spinosum layers, (2) corresponding fluorescence lifetime phasor plots, with gradient color distribution representing phase lifetime values, (3) Color-coded phasor-mapped images. **a Vitiligo:** In melanin-absent stratum corneum, keratin-rich areas are highlighted in yellow. The phasor distribution exhibits a comet-like structure pointing towards a fluorescence lifetime component of approximately 1.1 ns (yellow circle). Deeper in the epidermis, the phasor distribution shifts slightly toward shorter phase lifetimes. **b Skin type V:** Demonstrates a pronounced phasor shift due to highest melanin contribution, with the effect increasing with tissue depth. **c Skin type III:** Depth-resolved images and corresponding phasor plots demonstrate a gradual shift from longer fluorescence lifetimes in the stratum corneum toward longer lifetimes in the basal layer. Scale bars are 100 μm.

throughout the image (Fig. 5f). These images enable us to differentiate the keratin-filled horn cysts identified in the keratin map (red arrows) from melanin surrounding the dermal papilla (blue arrow) and other morphological structures exhibiting varying melanin-to-keratin ratios (violet arrows).

## Unmixing the protein-bound NADH component to capture cellular metabolic changes

We applied 4-component unmixing phasor analysis to the same in vivo images acquired using FLAME at various depths of the epidermis in subjects with skin types III, V, and vitiligo, in order to unmix the protein-bound

**Table 1 | Summary of fluorescence lifetime values used in the unmixing phasor analysis**

| Species | Lifetime (ns) | Source/Measurement Notes |
|---|---|---|
| Free NADH | 0.4 ns | Measured in solution (this work); consistent with refs. 29–32 |
| Protein-bound NADH | 3.4 ns | Measured in solution (literature values refs. 30–32) |
| Keratin | 1.1 ns | Estimated from in vivo imaging and phasor trajectory (this work; stratum corneum, vitiligo subject) |
| Melanin | 0.0 ns | Estimated from in vivo imaging and phasor trajectory (this work; basal keratinocytes, skin type V subject) |

NADH component (Fig. 6). Fig. 6a, c presents representative depth-resolved images acquired from these subjects, along with protein-bound NADH maps that illustrate the distribution of this molecular component throughout the images.

Similar to our approach for keratin and melanin, we used photon-weighted violin plots of the unmixed component fractions to capture variations in the contribution of protein-bound NADH across pixels. In these violin plots, the y -axis represents photon-weighted fractional intensity values, defined as the ratio of fluorescence from protein-bound NADH to the total NADH fluorescence at each pixel.

The protein-bound NADH distribution maps (Fig. 6c) reveal reduced fraction of protein-bound NADH in skin folds and in melanin-rich basal keratinocytes, while indicating a higher fraction in non-pigmented keratinocytes. The photon-weighted violin plots of the unmixed component fractions (Fig. 6d–f, Supplementary Data 1) show that the mode of the distribution for protein-bound NADH ratio shifts to lower fractional intensity values with increasing depth in both skin types III and V. This suggests a lower contribution of protein-bound NADH in the deeper epidermal layers, consistent with the observation that the basal and para-basal layers are more glycolytic compared to the upper epidermal layers[37]. In contrast, in vitiligo, the protein-bound NADH distributions is maintained across all depths, reflecting a potential metabolic alteration of the keratinocytes in vitiligo lesions.

An example of the integration of protein-bound NADH with the distributions of keratin and melanin to generate imaging maps with these unmixed components can be found in Supplementary Fig. 3.

To further evaluate the sensitivity of our unmixing approach to metabolic shifts, we conducted an experiment using freshly excised human skin tissue that was discarded during surgery. FLAME imaging was performed immediately after excision and again 6 hours later at the same location, with the tissue maintained at room temperature (Fig. 7). The phasor mapped images (Fig. 7c) generated from the color selections in the phasor plot (Fig. 7b) demonstrated a shift toward shorter fluorescence lifetimes after 6 hours.

To demonstrate that this shift is related to changes in the metabolic rate of keratinocytes reflected in the NADH fluorescence signal, we applied 4-component unmixing phasor analysis to these images, using the known phasor locations for keratin, melanin, protein-bound and free NADH. This analysis allowed us to generate fractional intensity maps that illustrate the distribution of protein-bound and free NADH throughout the images (Fig. 7d). These maps showed a decrease in the protein-bound NADH and an increase in free NADH after 6 hours. We used photon-weighted violin plots of the unmixed component fractions to capture variations in the contribution of these fluorescence species across pixels in these images (Fig. 7e, Supplementary Data 2). The violin plots revealed that the mode of the protein-bound NADH ratio distribution shifted to lower fractional intensity values after 6 hours, indicating an increase in free NADH characteristic of a glycolytic metabolic state[38]. The violin plots were generated after masking out keratin-rich large-scale features (primarily hairs) in the images to prevent these elements from dominating the analysis. Ex vivo conditions inevitably induce hypoxia and pH changes. The lifetime shifts we observed in excised skin tissue are consistent with published findings that hypoxia shortens NADH lifetimes and decreases the protein-bound fraction in cells and tissue slices[39,40]. Although not intended to replicate physiological metabolism, this experiment demonstrates the sensitivity of our approach to metabolic changes.

## Discussion

In this study, we used in vivo time-resolved TPEF imaging and advanced fluorescence lifetime phasor analysis to explore the contributions of key molecular components – specifically keratin, melanin, and both protein-bound and free NADH – to fluorescence signals in human skin. This innovative approach allows us to break down complex endogenous fluorescence signals, enhancing our ability to distinguish between molecular components, which is essential for advancing clinical multiphoton microscopy and improving noninvasive differential diagnoses and therapeutic evaluations.

The phasor approach to fluorescence lifetime enables the quantification and spatial distribution analysis of multiple fluorescent components within the same pixel through linear combinations of phasor representations derived from fluorescence decay curves[20]. This analysis relies on established phasor locations for specific molecular species. In our study, we utilized fluorescence phasor positions for specific molecular species in skin, using both literature data and our own in vivo TPEF measurements, thus enabling their characterization in the context of their natural skin environment.

To determine the keratin phasor position, we examined vitiligo lesions, where the absence of melanin in the stratum corneum allows keratin to emerge as the predominant fluorescent contributor. The phasor distribution observed in the stratum corneum pointed towards a fluorescence lifetime value around 1.1 ns, aligning with previously reported fluorescence lifetime keratin contribution in keratinocytes[35]. We have made the assumption that keratin is a single exponential with lifetime of 1.1 ns although it has been suggested that keratin exhibits a multi-exponential decay[36]. In other skin types, residual contributions from NADH or remnants of melanin can be present. While NADH concentrations may be diminished in the stratum corneum because of the dead nature of corneocytes, trace levels can persist from the underlying living epidermal layers. In contrast, the phasor distributions for pigmented normal skin types III and V exhibited a shift toward shorter fluorescence lifetimes, a change attributed to the contributions of melanin present in both the stratum corneum and deeper epidermal layers.

To define the melanin phasor position, we analyzed images from a volunteer with skin type V, where melanin is the primary fluorescent contributor. The phasor distribution in these melanin-rich areas showed a signature close to zero on the phasor plot, consistent with previous measurements[14,27,41], thus reinforcing the understanding of melanin's role in skin fluorescence. Notably, in the stratum corneum of both skin types III and V, we observed slight shifts toward shorter fluorescence lifetimes associated with melanin – potentially indicative of degraded melanin granules, despite some literature suggesting that such shifts may result from artifacts in histochemical staining[42,43].

The identification of phasor positions for melanin and keratin in human skin enabled us to effectively unmix these components from other fluorescent species, such as free and protein-bound NADH. By employing a four-component unmixing strategy[17–21], targeting free and protein-bound NADH, keratin, and melanin, we quantified the spatial distributions of keratin and melanin across various epidermal depths in subjects

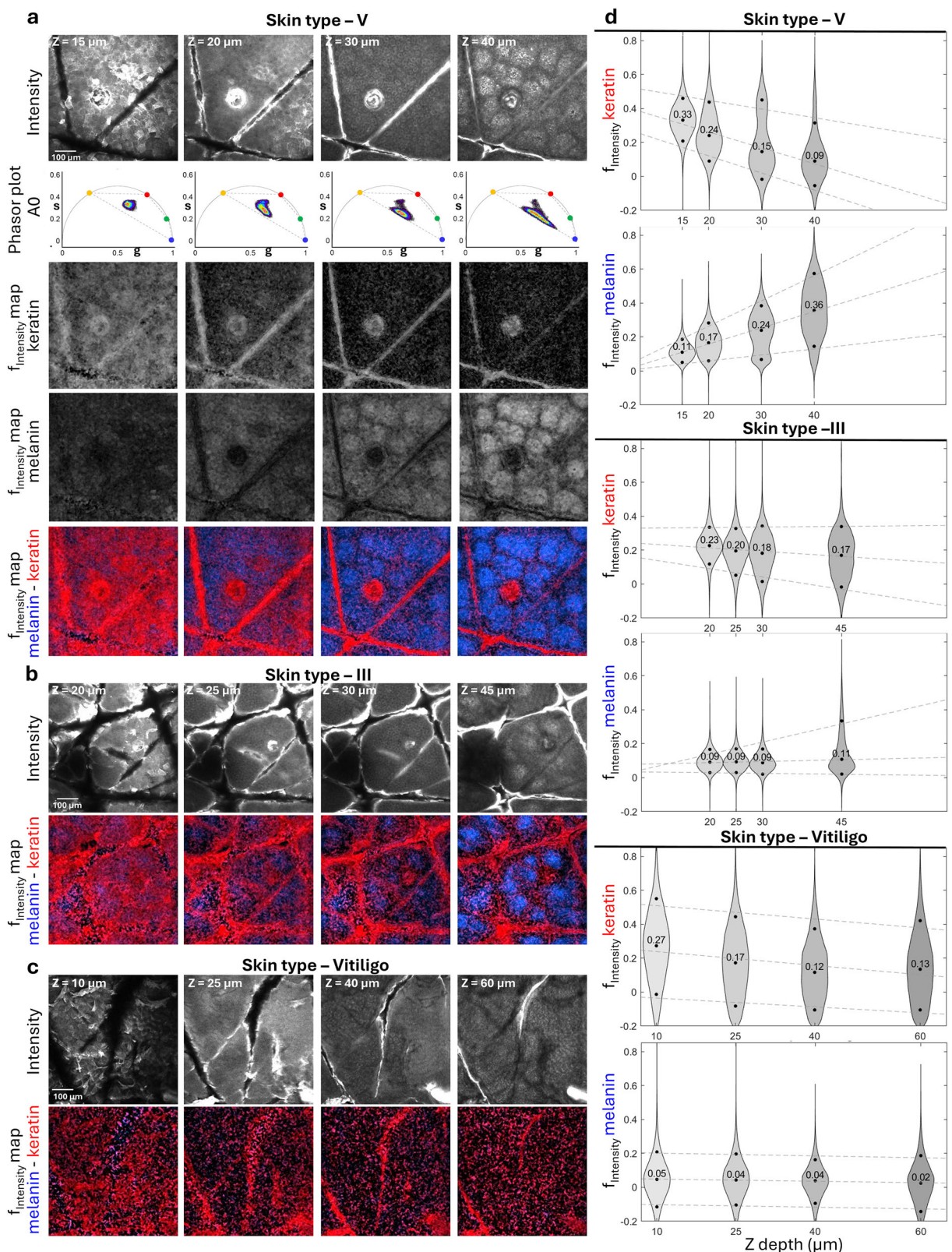

representative of different skin types. The reliability of this unmixing process was demonstrated by confirming the expected presence and variations of melanin and keratin across different depths and skin types, including vitiligo.

While previous studies have described approaches for melanin quantification in human skin using in vivo FLIM imaging[14] and phasor analysis[27],

the present study represents, to our knowledge, the first demonstration of unmixing endogenous skin components such as melanin and keratin, enabling visualization of distinct biochemical and structural features within a lesion. To illustrate the clinical applicability of our unmixing technique, we applied it to a freshly excised biopsy subsequently diagnosed as macular seborrheic keratosis (SK). The fractional intensity distribution maps

**Fig. 4 | Keratin and melanin unmixing reveals depth-dependent contributions across different Skin Types.** Representative data from three volunteers with varying melanin levels: skin type V (**a**), skin type III (**b**), and vitiligo (**c**) acquired in A0 channel. **a Skin type V:** Rows display: (1) depth-resolved intensity images, (2) shows the corresponding phasor plots, including the phasor position of the 4 components used in the multicomponent analysis: keratin (1.1 ns, red circle), melanin (0 ns, blue circle), protein-bound NADH (3.4 ns, yellow circle), and free NADH (0.4 ns, green circle). Dotted lines connect these positions, and any pixels contributing to these species are enclosed within this quadrilateral, (3-5) Fractional intensity maps for keratin, melanin, and a combined color-coded map, respectively. **b Skin type III:** Depth-resolved intensity images (top row) and color-coded fractional intensity maps for keratin and melanin (bottom row). **c Vitiligo:** Depth-resolved intensity

images (top row) and color-coded fractional intensity maps for keratin and melanin (bottom row). Note that deeper imaging was necessary to reach the dermal-epidermal junction due to increased thickness of the epidermis in this skin condition. Scale bars are 100 μm. **d Quantitative analysis of the unmixing**; Violin plots show fractional intensity ($f_{intensity}$) distributions of the unmixed pixel values of keratin and melanin for skin types V (top), III (middle), and vitiligo (bottom) across epidermal depths. Each violin plot illustrates the distribution of fractional intensity values for unmixed keratin and melanin components weighted by photons collected in each pixel. Percentile values [0.1, 0.5 (median), 0.9] for each distribution are displayed as black circles, and dashed lines are linear regression fits through these percentiles across depth to reveal depth-dependent trends. The intensities of the original images were corrected using an intensity histogram for visualization.

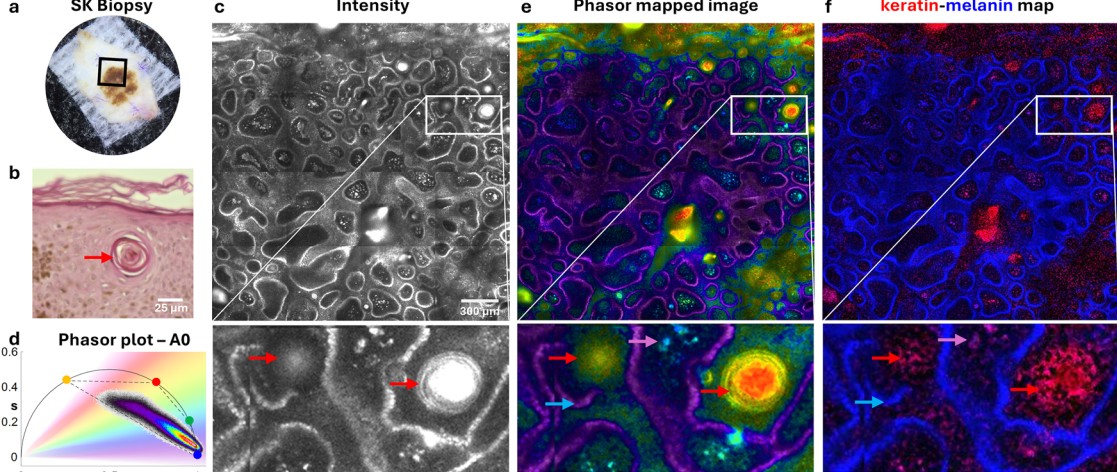

**Fig. 5 | Applying nnmixing of keratin and melanin for the detection of Seborrheic Keratosis (SK). a** Clinical image of the SK lesion. **b** H&E-stained biopsy section showing a horn cyst (red arrow) surrounded by melanin-loaded keratinocytes (brown). Scale bar 25 μm. **c** Intensity image acquired at the dermal-epidermal junction (z = 65 μm) using FLAME in time-resolved detection mode. Inset highlights horn cysts. **d** Corresponding lifetime phasor plots with gradient color distribution representing phase lifetime values. **e** Phasor-colored image showing the

distribution of phase lifetimes in the image. **f** Combined color-coded maps from 4-component unmixing phasor analysis, showing keratin (red) and melanin (blue) distributions. This analysis distinctly identifies keratin-filled horn cysts (red arrows) from the surrounding melanin-rich keratinocytes (blue arrows) and highlights other morphological structures that exhibit varying melanin-to-keratin ratios (indicated by violet arrows). Scale bar is 300 μm.

generated through phasor analysis distinguished critical features, such as keratin-filled "horn cysts" from other structures exhibiting varying melanin-to-keratin ratios. The ability to identify these key structural features label-free demonstrates the potential of this unmixing approach for distinguishing benign pigmented lesions from melanoma and for advancing real-time optical biopsy applications.

Finally, applying the four-component unmixing phasor analysis to assess the distribution of protein-bound NADH in in vivo images revealed decreased concentrations of protein-bound NADH in the deeper epidermal layers, highlighting the glycolytic nature of these basal layers. This finding aligns with results from a previous study that used a commercial MPM imaging platform, which achieved higher spatial resolution but limited scanning speed[37]. In contrast, depth analysis in vitiligo lesions showed stable levels of protein-bound NADH across all depths, suggesting notable metabolic alterations consistent with prior research[44].

In further experiments with freshly excised skin, we observed a shift towards shorter fluorescence lifetimes after six hours, correlating with a decrease in protein-bound NADH and an increase in free NADH, indicative of a glycolytic metabolic state[38]. These findings underscore the utility of phasor analysis in monitoring cellular metabolism and understanding skin health dynamics.

We implemented this analytical approach for the first time in a clinical setting using FLAME, our advanced imaging platform optimized for skin imaging[8]. This integration of effective in vivo imaging with comprehensive molecular analysis establishes FLAME as a powerful tool for non-invasive

clinical skin imaging with enhanced label-free molecular contrast. Compared to other label-free dermatologic imaging modalities, such RCM, OCT, line field confocal OCT (LC-OCT), MPM offers distinct advantages in combining submicron resolution with intrinsic molecular contrast derived from endogenous fluorophores. RCM enables rapid, wide-field en-face imaging with sub-cellular resolution and is widely used in clinical screening, but its contrast originates primarily from refractive index variations, limiting molecular specificity[45]. OCT provides deeper penetration (up to 1–2 mm) and quantitative structural information, yet it lacks cellular-level resolution[46]. LC-OCT enables rapid, wide-field cross-sectional imaging and sub-cellular resolution, but the absence of molecular specificity limits its use primarily to morphological assessments[47]. In contrast, MPM provides optical sectioning and molecular-level information through TPEF and SHG signals, allowing the visualization of biochemical and architectural features with histology-like, submicron detail. Current limitations of MPM include slower imaging speed and higher system complexity compared to RCM and OCT; however, ongoing developments in fast scanning approaches, compact femtosecond lasers, and phasor-based analysis are rapidly improving its translational potential. Collectively, these complementary characteristics position MPM as a molecularly informative extension to existing imaging modalities, offering unique opportunities for real-time, non-invasive optical biopsy and longitudinal monitoring of skin disease[48].

In conclusion, this exploratory study demonstrates the potential of advanced imaging and phasor analysis to improve label-free molecular contrast in clinical skin imaging. While the dataset is limited, these findings

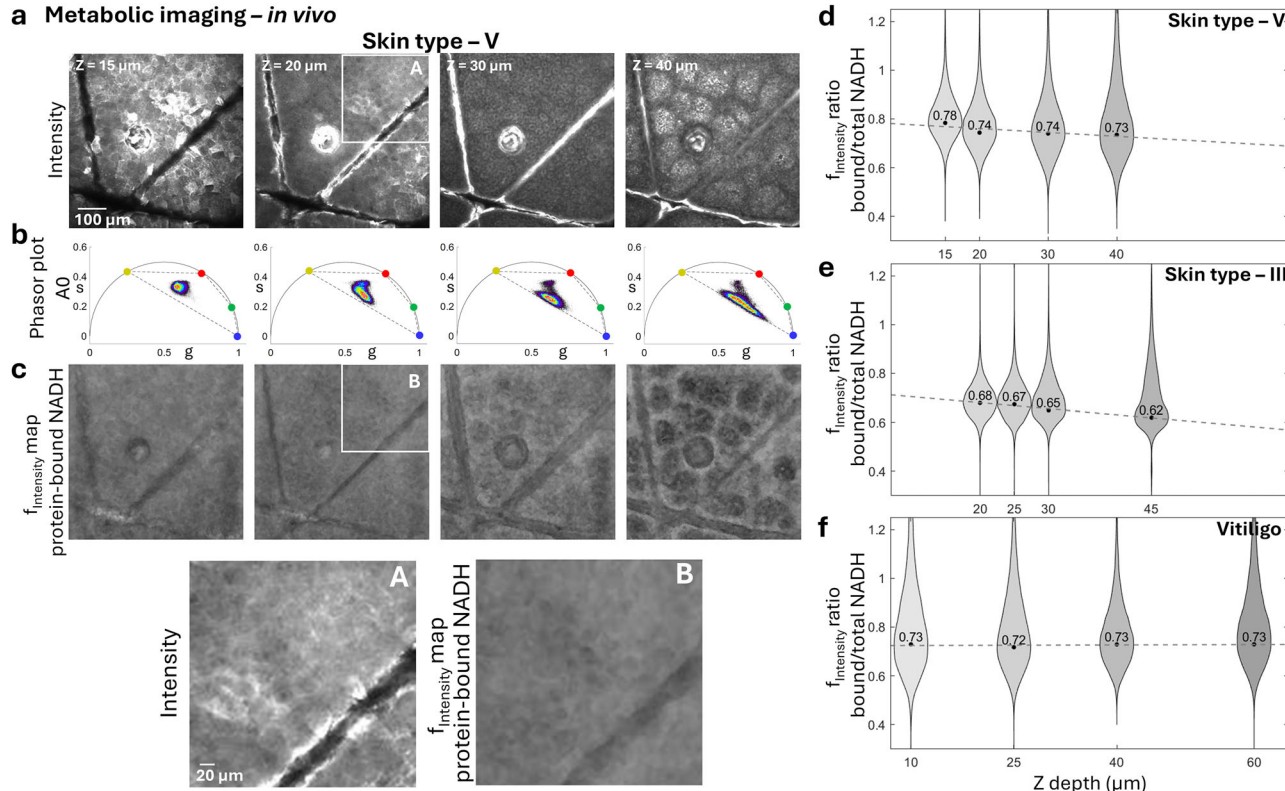

**Fig. 6 | Depth-dependent contribution of protein-bound NADH to phasor distribution in human Skin (in vivo).** Representative analysis from skin type V showing **a** depth-dependent intensity images, **b** corresponding fluorescence lifetime phasor plots in A0 channel, and **c** unmixed protein-bound NADH fractional intensity maps. (A,B) Insets highlight subcellular features in the unmixing. Scale bars are 100 μm and 20 μm. Photon-weighted violin plots of the unmixed component fractions corresponding to protein-bound to total NADH ratios for **skin type V** (**d**), **skin type III** (**e**) and **vitiligo** (**f**) across skin depth. Mode values for each distribution are displayed with black dots, and dashed lines represent linear fits to these values versus depth. In skin types III and V, the mode of the NADH ratio distribution shifts toward lower fractional intensity values with increasing depth, consistent with the higher glycolytic activity in the basal and para-basal layers compared to the upper epidermal layers. In contrast, vitiligo maintains consistent the protein-bound NADH distributions maintain consistent mode values across all depths, indicating potential metabolic alterations in the keratinocytes of vitiligo lesions.

provide a foundation for future larger-scale studies to assess the clinical utility of this approach.

## Methods

### Patients

The study enrolled two volunteers with normal skin, classified as Fitzpatrick skin types III and V. Additionally, we included a patient with vitiligo and another with a pigmented lesion. The pigmented lesion underwent a biopsy as per standard care procedures and was histopathologically diagnosed as macular seborrheic keratosis. Imaging sites included the forearm for the normal skin volunteers and the thigh lesion for the vitiligo patient. For logistical reasons, the seborrheic keratosis lesion, located on the patient's shoulder, was imaged ex vivo; this was necessary because the patient was seen at a clinic that did not have our FLAME instrument on site. The ages of the subjects that were imaged in vivo were between 34-40 years. All experiments were conducted with the full consent of each subject under a protocol approved by the Institutional Review Board for clinical research in human subjects at University of California, Irvine (HS# 2008-6307). All ethical regulations relevant to human research participants were followed.

### Clinical imaging platform (FLAME)

Clinical imaging was performed by using the fast, large-area multiphoton exoscope (FLAME), a portable imaging device based on multiphoton microscopy. This device, recently developed by our group, is highly optimized for clinical skin imaging[8]. It uses a turn-key femtosecond laser (Carmel 780, < 90 fs, 80 MHz, 780 nm excitation, Calmar, Palo Alto, CA)

and features an articulated arm attached to the imaging head, which houses near-infrared (NIR) optics, including a 4 kHz resonant-galvo beam scanning module, custom-designed relay, beam-expander optics and a 25X, 1.05 NA objective lens (XLPL25XWMP, Evident Scientific, Waltham, MA).

The optical design is optimized for sub-micron resolution, providing single field-of-view images of up to ~ 0.8 × 0.8 mm² at a rapid rate of 7.5 frames per second for ~ 1024 × 1024 pixels frame. During time-resolved fluorescence image acquisition, we typically accumulate between 20 and 30 frames, resulting in an effective acquisition time of 3 to 4 seconds per frame. For enhanced accuracy in this study, we increased the effective acquisition time to 9 seconds per 1200 × 1200 pixels image. The imaging area can be extended to several squared millimeter using a tile mosaic approach, where adjacent fields of view are stitched together, as detailed in a prior publication[8]. This imaging mode is facilitated by a motorized linear XY stage (Zaber Technologies) integrated into the FLAME imaging platform. The stage is attached to the imaging head and skin interface system, and it is automatically controlled to gently pull the skin during in vivo imaging, which allows for the acquisition and stitching of adjacent skin areas.

FLAME uses two hybrid photodetectors (R11322U-40-01, Hamamatsu, Bridgewater, NJ) for simultaneous signal acquisition. Our data acquisition card multiplies the 80 MHz laser clock by a factor of 32 and the analog output from the photodetectors is digitized at a rate of 2.7 GHz, achieving a temporal resolution of 0.39 ns for the fluorescence signal, closely matching the 0.4 ns rise time of the photodetectors.

In this study, we focused on the two-photon excited fluorescence (TPEF) detection, which was spectrally split into two channels by a dichroic

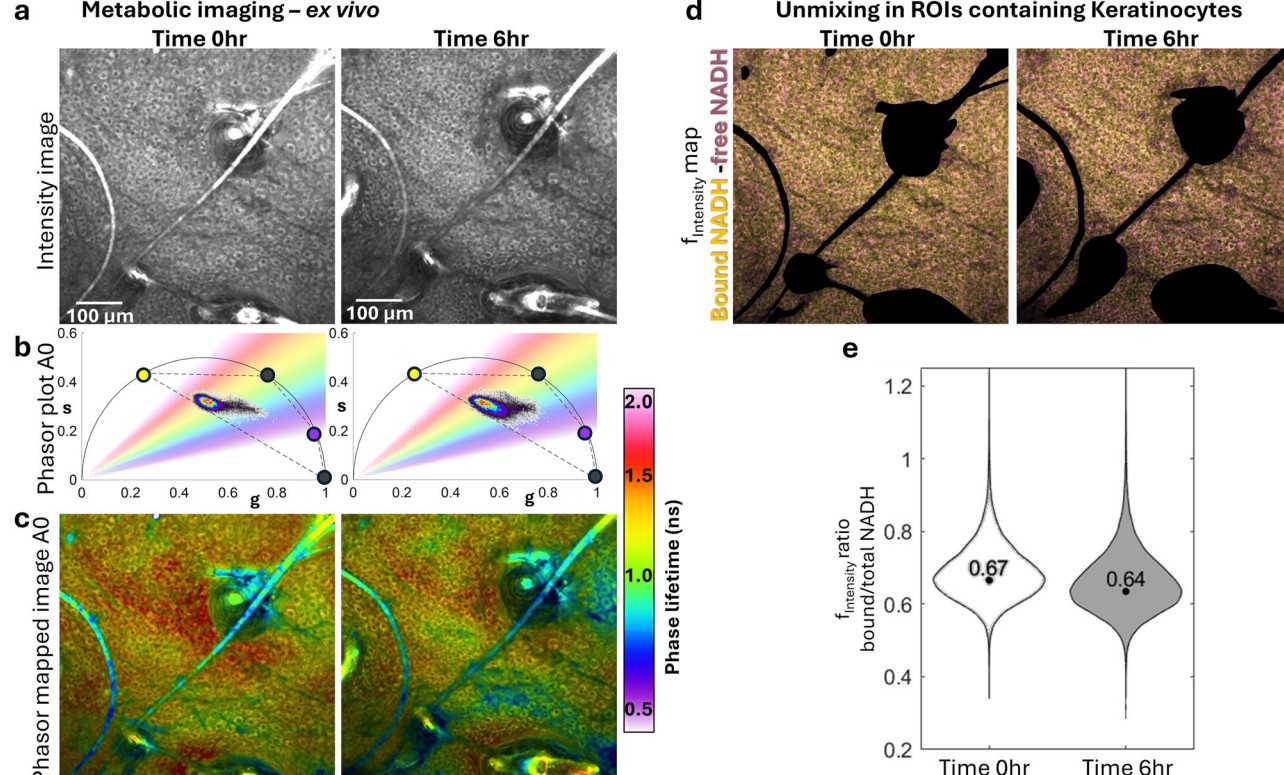

**Fig. 7 | Cellular metabolic changes in human skin (ex vivo) captured by unmixing the free and protein-bound NADH from keratin and melanin. a** Intensity images of epidermal keratinocytes in the human skin specimen at time 0 and after 6 hours at room temperature. **b** Corresponding fluorescence lifetime phasor plot in A0 channel, with a gradient color distribution representing phase lifetime values. **c** phasor mapped images showing a shift toward shorter fluorescence lifetimes after 6 hours. **d** Intensity images from (**a**) with keratin-rich large-scale features (primarily hairs) masked out; Color-coded maps combining the fractional intensities of protein-bound NADH (yellow) and free NADH (purple); **e** Photon-weighted violin plots of the unmixed component fractions showing that the mode of the protein-bound NADH distribution shifted to lower fractional intensity ratio values after 6 hours compared to time 0, indicating an increase in free NADH consistent to a glycolytic metabolic state. Scale bar are 100 μm.

mirror with a 506 nm cutoff wavelength (FF506-Di03, IDEX Health & Science, Northbrook, IL). One channel used a long pass filter (BLP01-405R, IDEX Health & Science) to maximize NADH fluorescence detection while minimizing FAD detection. The other channel used a bandpass filter (FF01-535/150, IDEX Health & Science) to detect fluorescence from all endogenous skin fluorophores.

Regarding laser power, our FLAME system uses an NA = 1.05 objective and a 60 mW excitation laser power to image beneath the skin surface. This is below the DNA and thermal damage thresholds established for two-photon microscopy of human skin[49,50]. Notably, FLAME operates with significantly lower laser fluence (1.7 times) and faster imaging time per unit area (40 times) compared to the values used for establishing the laser power threshold for two-photon microscopy on human skin[49,50].

### Image acquisition
Image acquisition for the FLAME device is controlled by using ScanImage software (MBF Bioscience, Ashburn, VA). In this study, individual images were acquired as 1200 ×1200 pixel frames over 600 μm x 600 μm skin area. For larger areas, we used the tile mosaic approach, to acquire images over 2.4 × 2.4 mm² as 4800 × 4800 pixel images. Coumarin 6 in ethanol (mono-exponential lifetime 2.5 ns) and Dimethyl POPOP in ethanol (mono-exponential lifetime 1.45 ns) (both from Fisher Scientific, Waltham, MA) were used for the phasor plot calibration.

### Image analysis
The advanced phasor analysis in this study was performed by using an open-source software platform, GSLab, recently developed in our lab[22]. Calibration standards, imaged in each session, are used to compute the modulation

factor and phase shift, which account for the instrument response and experimental variability[51]. The phasor transform of the temporal fluorescence signal at each pixel is computed through Fourier decomposition, specifically extracting the first harmonic's two coefficients to define a coordinate pair for each pixel. These coordinates are then plotted on a phasor plot, a 2-dimensional histogram representing pixel density[52]. To determine the fluorescence lifetime for each pixel, the corresponding phasor coordinates are projected onto the universal circle, identifying the point with the same phase shift as the measured signal. This phase-lifetime corresponds to the decay time of a single-exponential species with the same phase shift as the fluorescence measurement. The projection is visualized in the figures as a fan-shaped color gradient on the phasor plot, which encodes lifetime values in the fluorescence image based on each pixel's phasor position. We utilized fluorescence phasor positions for specific molecular species in skin, using both literature data and our own in vivo TPEF measurements. Using these values and the phasor positions in the first two harmonics, we perform pixel-wise component unmixing[20]. For each pixel, the algorithm calculates the number of photons contributed by each of these four components based on the pixel's intensity. The result is a fractional set of images, one for each component, where each pixel represents the associated photon fraction for that component. These images can then be color-coded into a merged image representing the direct sum of the individual components. Additionally, photon-weighted violin plots of the unmixed component fractions were generated, with each pixel contributing based on its intensity. For better visualization, the intensity of each figure is normalized individually.

In the analysis pipeline, raw data were loaded into GSLab to generate the intensity image (Supplementary Fig. 4a; 1024 × 1024 pixels) and the corresponding phasor plot (Supplementary Fig. 4b). Two types of phasor-

mapped images were produced: (1) a phasor map showing color distribution (Supplementary Figs, 4c) and (2) a phasor color map overlaid on the intensity image (Supplementary Fig. 4d), which was the focus of this study. A similar approach was employed for component mapping, where the intensity underlay represents the total component contribution per pixel, and the phasor colors indicate the relative proportions of these components. Consequently, the phasor-mapped images exhibit characteristics similar to those of the intensity images, as discussed in the Results section.

The typical photon count per pixel was ~200–250. In multicomponent scenarios, phasor coordinates can spread beyond the tetragon defined by the four reference components, lowering the signal-to-noise ratio (SNR). To improve SNR, the phasor data were downsampled to $256 \times 256$ pixels (Supplementary Fig. 4e), effectively binning neighboring pixels to increase photon counts without discarding information. A $3 \times 3$ median filter was applied in the phasor domain (Supplementary Fig. 4f) to further reduce spread, while leaving the spatial intensity images unaffected (Supplementary Fig. 4c,g). As a result, the intensity-overlaid phasor images retain their visual characteristics before and after processing.

## Component selection and deconvolution strategy

The A1 image was used exclusively to determine the phasor position of melanin, due to its high abundance in A1 and in skin type V. All subsequent deconvolution and fractional intensity calculations were performed using the A0 image. This choice was motivated by the reduced number of contributing components in A0, which simplifies the unmixing process and improves robustness. Notably, FAD and lipofuscin, if present, do not contribute significantly to A0, allowing for a clearer mapping of the remaining components. Although the high signal-to-noise ratio (SNR) in the acquired data theoretically permits deconvolution of all components, this is often not feasible in human in vivo imaging due to limited signal strength.

## Data availability

The datasets generated and/or analyzed in this study are available from the corresponding authors upon request. Source data underlying graphs 4 and 6 can be obtained from Supplementary data 1, and source data underlying graph 7 can be obtained from Supplementary data 2.

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

## Acknowledgements

We acknowledge the funding support for this project from the NIBIB (R01EB026705), NIAMS (R21AR082648) and NCI (R01CA259019), as well as from the Skin Biology Resource-Based Center (P30AR075047) at the UC Irvine. The Nonlinear Optical Microscopy Lab conducting this study is part of the Optical Biology Core Facility, a shared resource supported by the Chao Family Comprehensive Cancer Center (P30CA062203) at the UC Irvine. SR also acknowledges support from NIGMS (1R35GM154815) and NCI (R01CA279195).

## Author contributions

Conceptualization and study design: S.R., B.T. M.B. Development and optimization of the instrumentation for in vivo imaging: A.D., A.F.D., A.V. Data collection: S.R., B.T. Development of methodology: S.R., B.T., M.B. Data analysis and interpretation of data: S.R., B.T., A.V. Programming: A.V. Biopsy acquisition and clinical insight: A.G., K.M.K. Manuscript writing: S.R. and M.B. with input from all the authors. Manuscript review: all authors.

## Competing interests

The authors declare no competing interests.
