## [Transparent Peer Review file · Communications Biology]

Label-Free Human Skin Imaging with Enhanced Molecular Contrast via Time-Resolved Fluorescence and Advanced Phasor Analysis

Corresponding Author: Dr Suman Ranjit

Version 0:

Reviewer comments:

Reviewer #1

(Remarks to the Author)

This manuscript clearly demonstrates the clinical promise of label-free two-photon fluorescence lifetime imaging for dermatological applications. By combining depth-resolved acquisition with phasor analysis, the authors achieve selective readouts of endogenous fluorophores in skin like keratin, melanin, NADH and FAD which enabling mapping of biochemical composition in live skin. The work is a compelling example of translational microscopy and could be highly impactful for the biomedical imaging community. Nevertheless, several issues should be addressed before publication.

1.(a) The reference phasor positions were taken from earlier studies, yet the actual lifetime (or phasor) values are scattered through literature. Please provide a concise table listing the lifetimes or phasor coordinates that were adopted for each fluorophore.

(b) In addition, Fig. 1c shows free-FAD has a shorter lifetime and protein-bound FAD has a longer lifetime. This is the opposite of most reports. Please confirm these assignments.

2.Since the age of the subjects is not reported, it is difficult to evaluate possible lipofuscin interference. Because lipofuscin becomes prominent in aged skin and can overlap Channel A1, it would be good to discuss whether the cohort's age distribution could affect the results.

3.To quantify crosstalk among the four components, it would be helpful to image an artificial sample containing pure solutions of free NADH, protein-bound NADH, keratin and melanin under comparable photon counts. Reporting the error rates of the four component unmixing on such a control would strengthen confidence of the in vivo data.

4.The metabolic shift seen in Fig. 6 seems to align with the glycolysis to OXPHOS transition reported in Ref. 41 for vitiligo lesions. Did the FAD channel exhibit a complementary lifetime or intensity shift indicative of OXPHOS activation? Please also consider presenting the optical redox ratio $[FAD/(FAD + NADH)]$ and a lifetime based redox metric (bound NADH fraction / bound FAD fraction) to reinforce the metabolic interpretation.

5.Figure 7 suggests a glycolytic shift in the NADH channel over 6 h. How did the FAD channel behave over the same period—unchanged or shifted? A quantitative side by side plot of NADH and FAD lifetime or intensity versus time would clarify whether both cofactors report consistent metabolic dynamics.

6.The authors used a pixel-wise 4-component unmixing method to multiplex free-NADH, bound-NADH, keratin, and melanin. This is a smart approach to disentangling a complex biological system. However, one important point needs further discussion. In this unmixing strategy, each component appears to be considered as a single-exponential decay, as indicated by their positions on the phasor semi-circle. But, as the authors themselves note, keratin exhibits a multi-exponential decay profile. This assumption could potentially distort the phasor location not only in the first harmonic, but also in higher harmonics thereby introducing significant error. The authors should clarify whether this single-exponential assumption is justified and provide an estimate of the error it might introduce.

7.Given that we are looking at autofluorescence from live tissue, where collecting a sufficient number of photons is

challenging. The phasor histogram appears to exhibit remarkably low shot noise. How many photons per pixel were typically collected? Additionally, was a median filter applied prior to phasor plot? If so, what was the kernel size?

Minor comment:

1. All phasor histogram plots are shown at low image resolution, making it difficult to discern details. Please consider replacing them with higher-resolution figures.
2. In Figure 6 (d), (e), and (f), it would be helpful to indicate statistical significance across different depths for each tissue type.

Reviewer #2

(Remarks to the Author)

It was a pleasure to review your manuscript, "Label-Free Human Skin Imaging: Enhanced Molecular Contrast via Time-Resolved Fluorescence and Advanced Phasor Analysis," and I am pleased to share my feedback with you.

Summary:

This manuscript introduces a clinical workflow that integrates the FLAME (fast large-area multiphoton exoscope) system with multi-harmonic phasor analysis to achieve in-situ, four-component unmixing of endogenous skin fluorophores, including keratin, melanin, free NADH, and protein-bound NADH. Notably, the manuscript aims to address long standing technical challenges in label free multiphoton microscopy (MPM) by combining fast hardware, a large field of view (FOV), and advanced lifetime unmixing strategies. In the conclusion, the authors state that this study lays the groundwork for future research focused on real-time, precise assessment of skin disorders and the monitoring of therapeutic responses.

Comments:

Overall, the manuscript presents a novel application of phasor-based unmixing in clinical skin imaging. However, several critical issues must be addressed, particularly regarding validation, statistical robustness, and biological interpretation, before the manuscript can be considered for publication.

1. Validation of Phasor Component Positions:

The phasor positions for key fluorophores are derived from single images or literature values (e.g., keratin at 1.1 ns, NADH at 3.4 ns). However, variable values are available for NADH and keratin in the literature. In addition, Free NADH, for example, exists in at least two conformational states ("folded" and "extended") each with distinct fluorescence lifetimes. This conformational heterogeneity can produce bi-exponential decay profiles even in the absence of protein binding. In figure 1, free and protein bound NADH located on the unit circle in the phasor plot. Any difference in the phasor position of reference can lead to the inaccurate unmixing. Therefore, validation is required through phasor plots of pure (free) and protein bound NADH (e.g., with ADH or MDH) solutions. Similarly, independent phasor diagrams for pure keratin and melanin solutions should be included.

2. Limited Sample Size and Lack of Statistical Validation:

The study includes only three healthy volunteers, one vitiligo case, and one seborrheic keratosis (SK) biopsy. No quantitative statistical analyses are provided for the presented data. I recommend either expanding the dataset to include more samples with appropriate statistical evaluation or rephrasing the conclusions to emphasize the exploratory nature of the study and outlining future directions in the introduction/discussion.

3. Interpretation of Metabolic Changes:

The manuscript discusses cellular metabolic changes based on the unmixing of free and protein-bound NADH. However, tissue maintained ex vivo for up to 6 hours at room temperature is likely to undergo hypoxia and pH changes, all of which can shift metabolism toward glycolysis. These ex vivo artifacts make it challenging to interpret lifetime shifts as genuine metabolic changes. Supporting evidence from biochemical assays or literature clearly linking lifetime parameters to metabolic pathways and cellular viability should be included.

4. Missing Comparative Analysis:

Figure 5 lacks a comparison between phasor-FLIM and conventional diagnostic methods. Including histopathological analysis of the biopsy would strengthen the manuscript by providing a reference standard for comparison.

5. Incorrect Phasor Positions in Figure 1:

The positions of free and bound FAD are incorrect. Free FAD has a longer lifetime and should be located to the left side of the phasor plot, whereas protein-bound FAD should appear on the right. Additionally, the phasor position of SHG is not

included in the figure and should be added.

Minor Comments:

State the reason for using a single excitation wavelength for all data acquisition.

Did you attempt to include FAD as a fifth fluorophore? If so, please explain why it was ultimately excluded.

In Figure 3, the interpretation that phasor contributions represent keratin is unclear. A "comet-like" structure indicating keratin is not consistently visible across all phasor plots, unlike melanin. Please clarify.

In Figure 4.e, why do keratin and melanin distributions remain unchanged for skin type 3?

In Figure 6, there appears to be no change in NADH fractions across depths except at 45 μm for skin type 3. Could this be due to spectral bleed-through from keratin or melanin into the NADH signal?

Phasor plots across figures are densely packed and difficult to interpret. Please improve plot visibility for clarity.

Briefly outline the diagnostic limitations of current label-free dermatologic imaging modalities (such as confocal microscopy, RCM, and OCT) and explain why molecular specificity is important. Additionally, consider including a comparison of the strengths and limitations (both technical and clinical) of FLAME relative to these conventional imaging techniques in the introduction or discussion section.

Reviewer #3

(Remarks to the Author)
Please see attached document.

Version 1:

Reviewer comments:

Reviewer #1

(Remarks to the Author)
The authors have thoroughly addressed all previous comments and provided clear clarifications to the reviewers' concerns. I have no additional comments on the revised version. The manuscript is well-prepared, and I recommend it for publication.

Reviewer #2

(Remarks to the Author)
The authors have substantially revised their manuscript in response to my previous comments. The study remains an important contribution to the field of label-free dermatological imaging, demonstrating the translational potential of time-resolved multiphoton microscopy combined with phasor-based lifetime analysis. The revisions have addressed most concerns from the previous round. However, a few remaining aspects related to validation and completeness of the reference phasor dataset should be clarified or acknowledged more explicitly.
The authors have now included Table 1 summarizing fluorescence lifetime values for free NADH, protein-bound NADH, keratin, and melanin. But the validation remains partially qualitative because the supporting phasor plots are not part of the manuscript.
So, I recommend acceptance after minor revision:

1. Including or referencing phasor plots (or g, s coordinates) for calibration species (NADH free, NADH bound, keratin and melanin) in the supplementary materials.
With minor additions, the paper will provide a complete and reproducible resource for researchers in clinical biophotonics and metabolic imaging.

Reviewer #3

(Remarks to the Author)
See attached file

We thank the reviewers for their thoughtful comments and suggestions. Please see our point-by-point responses to the comments and issues raised. The changes in the manuscript have been highlighted in red. We provide a list of references at the end of the responses to reviewers' comments. Please, note that these citations correspond to different numbers in the manuscript or have only been used for the responses to reviewers' comments.

Reviewer #1 (Remarks to the Author):

This manuscript clearly demonstrates the clinical promise of label-free two-photon fluorescence lifetime imaging for dermatological applications. By combining depth-resolved acquisition with phasor analysis, the authors achieve selective readouts of endogenous fluorophores in skin like keratin, melanin, NADH and FAD which enabling mapping of biochemical composition in live skin. The work is a compelling example of translational microscopy and could be highly impactful for the biomedical imaging community. Nevertheless, several issues should be addressed before publication.

Question 1(a): The reference phasor positions were taken from earlier studies, yet the actual lifetime (or phasor) values are scattered through literature. Please provide a concise table listing the lifetimes or phasor coordinates that were adopted for each fluorophore.

We thank the reviewer for this comment. Not all phasor positions were taken from the literature; to clarify, we now provide a summary table (Table 1) listing the fluorescence lifetimes adopted for each fluorophore. Free NADH, initially reported from the literature, is now based on our own solution measurements validated against published data (Figure 1 below – not included in the manuscript). Protein-bound NADH is taken from the literature, while keratin and melanin are determined through *in vivo* imaging to capture their physiologically relevant signatures. This table has been added to the revised manuscript in the Results section (p. 9)

Table 1. Summary of fluorescence lifetime values used in the unmixing phasor analysis

Species	Fluorescence lifetime (ns)	Source/Measurement Notes
Free NADH	0.4 ns	Measured in solution (this work); consistent with Refs ¹⁻⁴
Protein-bound NADH	3.4 ns	Measured in solution (literature values Refs) ²⁻⁴
Keratin	1.1 ns	Estimated from in vivo imaging and phasor trajectory (this work; stratum corneum, vitiligo subject)
Melanin	0.0 ns	Estimated from in vivo imaging and phasor trajectory (this work; basal keratinocytes, skin type V subject)

Note. Although not used in our unmixing phasor analysis, for additional validation, we also measured the fluorescence lifetime of free FAD as shown below along with the measurement of free NADH

Figure 1. Free NADH (3.5 mM) and FAD (100 μM) solutions in 200 mM Tris-HCl buffer exhibited phase lifetime values of 0.4 ns and 2.4 ns, respectively, when measured using FLAME. These values are consistent with previously reported literature values ¹⁻⁴

Question 1(b): In addition, Fig. 1c shows free-FAD has a shorter lifetime and protein-bound FAD has a longer lifetime. This is the opposite of most reports. Please confirm these assignments.

We thank the reviewer for noting this error, which has now been corrected in Fig. 1c.

Question 2: Since the age of the subjects is not reported, it is difficult to evaluate possible lipofuscin interference. Because lipofuscin becomes prominent in aged skin and can overlap Channel A1, it would be good to discuss whether the cohort's age distribution could affect the results.

The ages of the subjects were between 34-40 years. We added this information to the manuscript In Methods section-Patients.

Channel A1 was used solely to determine the phasor position of melanin, which is particularly abundant in channel A1 (506-610 nm) and in skin type V. However, all unmixing and fractional intensity calculations were performed using channel A0 (405-506 nm). Lipofuscin (emission spectrum peak around 600nm) does not significantly contribute to channel A0, which enhances the specificity of component mapping.

Question 3: To quantify crosstalk among the four components, it would be helpful to image an artificial sample containing pure solutions of free NADH, protein-bound NADH, keratin and melanin under comparable photon counts. Reporting the error rates of the four component unmixing on such a control would strengthen confidence of the *in vivo* data.

We thank the reviewer for the suggestion. While solution measurements can provide approximate error estimates, the value of our approach is in capturing the phasor signatures of endogenous skin fluorophores *in vivo*, within their native tissue environment. To obtain physiologically relevant fluorescence lifetimes for keratin and melanin, we performed *in vivo* imaging under conditions where each fluorophore is the primary contributor to the autofluorescence signal: keratin in the stratum corneum of a vitiligo subject (absence of melanin) and melanin in basal keratinocytes of a highly pigmented skin type V subject. This strategy allowed us to capture the intrinsic phasor signatures of these endogenous fluorophores in human skin tissue. We agree that additional control experiments with pure solutions could provide complementary error estimates, but they cannot substitute for tissue-based measurements that capture fluorescence behavior in the native biological context.

To better clarify our strategy and study rationale, we revised the opening of the Results section (p. 3), which now reads:

“Fluorescence lifetime detection was performed to analyze fluorescence lifetime properties of skin endogenous fluorophores across two spectral channels: A0 (405 - 506 nm) and A1 (506 - 610 nm). Two-photon excited fluorescence (TPEF) decays were transformed into phasor plots used to quantify fluorescence lifetimes and enhance molecular contrast by color-coding based on the measured phase lifetimes (Figure 1b).

Reference phasor positions for the major skin fluorophores were established by combining published data with our own measurements (Figure 1c). Free NADH and free FAD were measured in solution with our system and validated against literature values,¹⁻⁴ while protein-bound NADH lifetime were adopted from published reports.²⁻⁴ In contrast, keratin and melanin lifetimes were determined *in vivo*, where specific physiological conditions allowed each to dominate the autofluorescence signal: keratin in the stratum corneum of a vitiligo subject (melanin absent) and melanin in basal keratinocytes of a highly pigmented skin type V subject. This strategy enabled us to obtain intrinsic phasor signatures of these fluorophores directly in human skin tissue, rather than relying solely on solution-based values. The two detection spectral windows, A0 and A1 as defined above, were selected based on the TPEF emission spectra of these molecular species (Figure 1d).⁵ For the unmixing analysis, however, we restricted detection to the A0 channel (405–506 nm), where FAD contribution is minimal, in order to reduce analysis complexity. Thus, while FAD was characterized for reference, it was not included in the unmixing analysis. Instead, we focused on separating four distinct components: free and protein-bound NADH from keratin and melanin (Figure 1e). We generated fractional intensity maps that provide spatial resolution of photon distributions associated with key cellular features and quantified their contributions. This approach leverages physiologically relevant *in vivo* lifetime signatures to achieve label-free molecular contrast in human skin, as described in the following sections.”

Question 4: The metabolic shift seen in Fig. 6 seems to align with the glycolysis to OXPHOS transition reported in Ref. 41 for vitiligo lesions. Did the FAD channel exhibit a complementary lifetime or intensity shift indicative of OXPHOS activation? Please also consider presenting the optical redox ratio $[FAD/(FAD + NADH)]$ and a lifetime based redox metric (bound NADH fraction / bound FAD fraction) to reinforce the metabolic interpretation.

Optical redox ratios and lifetime-based redox metrics can be measured in simplified systems such as cultured cells, where NADH (free and protein-bound) and FAD (free and protein-bound) are the primary fluorophores contributing to the 4-component unmixing analysis. In contrast, in epidermal keratinocytes in vivo, additional fluorophores such as keratin and melanin significantly contribute to the fluorescence signal, complicating straightforward calculation of these metrics. To address this complexity, our microscope detection channels were designed to perform phasor unmixing on TPEF signals detected in channel A0 (405–506 nm), where FAD contribution is minimal. This strategy allows robust separation of NADH, keratin, and melanin signals while maintaining physiologically relevant interpretation of NADH-related metabolic changes.

Question 5: Figure 7 suggests a glycolytic shift in the NADH channel over 6 h. How did the FAD channel behave over the same period—unchanged or shifted? A quantitative side by side plot of NADH and FAD lifetime or intensity versus time would clarify whether both cofactors report consistent metabolic dynamics.

As described above, the A0 channel (405–506 nm) was chosen to minimize FAD contribution in vivo, so reliable FAD lifetime or intensity measurements were not possible in this dataset.

Question 6: The authors used a pixel-wise 4-component unmixing method to multiplex free-NADH, bound-NADH, keratin, and melanin. This is a smart approach to disentangling a complex biological system. However, one important point needs further discussion. In this unmixing strategy, each component appears to be considered as a single-exponential decay, as indicated by their positions on the phasor semi-circle. But, as the authors themselves note, keratin exhibits a multi-exponential decay profile. This assumption could potentially distort the phasor location not only in the first harmonic, but also in higher harmonics thereby introducing significant error. The authors should clarify whether this single-exponential assumption is justified and provide an estimate of the error it might introduce.

We thank the reviewer for raising this important point. While keratin does appear to exhibit a multi-exponential decay profile, as also suggested in prior publications,⁶ our justification for using a single-exponential approximation was based on the fact that in images acquired from keratin-rich structures such as the stratum corneum, skin folds, and hairs, the phasor distribution consistently forms a characteristic “comet-like” trajectory converging toward ~1.1 ns, as also reported in previous studies.⁷ We therefore adopted this as the reference phasor position for keratin. We acknowledge that this simplification may introduce some error, but the impact appears minimal in practice. Importantly, when this reference is combined with melanin, free NADH, and protein-bound NADH in the 4-component unmixing, the analysis yields biologically consistent results, for example, keratin enrichment in skin folds and melanin localization in the basal layer (melanin caps), as shown in Fig. 4. These findings support both the validity and the practical utility of this approximation for resolving endogenous skin fluorophores in vivo.

Question 7: Given that we are looking at autofluorescence from live tissue, where a sufficient number of photons is challenging. The phasor histogram appears to exhibit remarkably low shot noise. How many photons per pixel were typically collected? Additionally, was a median filter applied prior to phasor plot? If so, what was the kernel size?

Raw data were processed in GSLab to generate intensity images (1024 × 1024 pixels) and corresponding phasor plots. We typically collected ~200–250 photons/pixel. In multicomponent scenarios, phasor coordinates can spread due to shot noise; to improve SNR, the data were downsampled to 256 × 256 pixels by binning neighboring pixels, which increases photon counts per pixel without discarding information. A 3 × 3 median filter was then applied in the phasor domain only, leaving the spatial intensity images unaffected. To clarify these steps, we added the following paragraph in the Methods/Image Analysis section and included Supplemental Figure 4, illustrating the raw data, downsampling, and phasor filtering.

“In the analysis pipeline, raw data were loaded into GSLab to generate the intensity image (Supplement Figure 4a; 1024 × 1024 pixels) and the corresponding phasor plot (Supplement Figure 4b). Two types of phasor-mapped images were produced: (1) a phasor map showing color distribution (Supplement Figure 4c) and (2) a phasor color map overlaid on the intensity image (Supplement Figure 4d), which was the focus of this study. A similar approach was employed for component mapping, where the intensity underlay represents the total component contribution per pixel, and the phasor colors indicate the relative proportions of these components. Consequently, the phasor-mapped images exhibit characteristics similar to those of the intensity images, as discussed in the Results section.

The typical photon count per pixel was $\sim 200\text{--}250$. In multicomponent scenarios, phasor coordinates can spread beyond the tetragon defined by the four reference components, lowering the signal-to-noise ratio (SNR). To improve SNR, the phasor data were downsampled to 256×256 pixels (Supplement Figure 4e), effectively binning neighboring pixels to increase photon counts without discarding information. A 3×3 median filter was applied in the phasor domain (Supplement Figure 4f) to further reduce spread, while leaving the spatial intensity images unaffected (Supplement Figure 4c,g). As a result, the intensity-overlaid phasor images retain their visual characteristics before and after processing.”

Supplemental Figure 4. Pipeline for FLIM data visualization and analysis. **a**, Intensity image displayed in GSLab at full resolution (1024×1024 pixels). **b**, Corresponding phasor plot without downsampling or filtering. **c**, Phasor-mapped image showing color-coded distribution in phasor space. **d**, Phasor-mapped image overlaid with the original intensity image. **e**, Reduced phasor spread following downsampling to 256×256 pixels. **f**, Application of a 3×3 median filter in phasor space to further reduce noise. **g**, Phasor-mapped image after downsampling and filtering. **h**, Final phasor-mapped image overlaid with the intensity image, corresponding to the original image in (a). Scale bar: $100 \mu\text{m}$.

Minor comments:

Minor comment 1. All phasor histogram plots are shown at low image resolution, making it difficult to discern details. Please consider replacing them with higher-resolution figures.

All the phasors are now replaced with high resolution images.

Minor comment 2. In Figure 6 (d), (e), and (f), it would be helpful to indicate statistical significance across different depths for each tissue type.

For the results shown in Figure 6, each region of interest (ROI) includes one image acquired at each depth, with no repeated measurements within the same ROI and therefore, statistical testing across depths is not meaningful. To address the reviewer’s comment, we revised Figure 6 to display the data as violin plots of the fractional intensity ratio (bound/total NADH), which better show the distribution and depth-dependent trends. These trends are now discussed in the text.

Reviewer #2 (Remarks to the Author):

It was a pleasure to review your manuscript, “Label-Free Human Skin Imaging: Enhanced Molecular Contrast via Time-Resolved Fluorescence and Advanced Phasor Analysis,” and I am pleased to share my feedback with you.
Summary:

This manuscript introduces a clinical workflow that integrates the FLAME (fast large-area multiphoton exoscope) system with multi-harmonic phasor analysis to achieve in-situ, four-component unmixing of endogenous skin fluorophores, including keratin, melanin, free NADH, and protein-bound NADH. Notably, the manuscript aims to address long standing technical challenges in label free multiphoton microscopy (MPM) by combining fast hardware, a large field of view (FOV), and advanced lifetime unmixing strategies. In the conclusion, the authors state that this study lays the groundwork for future research focused on real-time, precise assessment of skin disorders and the monitoring of therapeutic responses.

Comments:

Overall, the manuscript presents a novel application of phasor-based unmixing in clinical skin imaging. However, several critical issues must be addressed, particularly regarding validation, statistical robustness, and biological interpretation, before the manuscript can be considered for publication.

Question 1:

The phasor positions for key fluorophores are derived from single images or literature values (e.g., keratin at 1.1 ns, NADH at 3.4 ns). However, variable values are available for NADH and keratin in the literature. In addition, Free NADH, for example, exists in at least two conformational states (“folded” and “extended”) each with distinct fluorescence lifetimes. This conformational heterogeneity can produce bi-exponential decay profiles even in the absence of protein binding. In figure 1, free and protein bound NADH located on the unit circle in the phasor plot. Any difference in the phasor position of reference can lead to the inaccurate unmixing. Therefore, validation is required through phasor plots of pure (free) and protein bound NADH (e.g., with ADH or MDH) solutions. Similarly, independent phasor diagrams for pure keratin and melanin solutions should be included.

To clarify the phasor positions used in our analysis, we now provide a summary table (Table 1) listing the fluorescence lifetimes adopted for each fluorophore. Free NADH was measured in solution with our FLAME system and validated against published values, while protein-bound NADH lifetimes were taken from the literature. Keratin and melanin values were determined *in vivo*, where specific physiological conditions allowed each fluorophore to dominate the autofluorescence signal (keratin in the stratum corneum of a vitiligo subject and melanin in basal keratinocytes of a highly pigmented subject). This strategy ensured that the reference values reflect physiologically relevant phasor signatures. Table 1 has been added to the revised manuscript in the Results section (p. 9):

Table 1. Summary of fluorescence lifetime values used in the unmixing phasor analysis

Species	Fluorescence lifetime (ns)	Source/Measurement Notes
Free NADH	0.4 ns	Measured in solution (this work); consistent with Refs ¹⁻⁴
Protein-bound NADH	3.4 ns	Measured in solution (literature values Refs) ²⁻⁴
Keratin	1.1 ns	Estimated from in vivo imaging and phasor trajectory (this work; stratum corneum, vitiligo subject)
Melanin	1.0 ns	Estimated from in vivo imaging and phasor trajectory (this work; basal keratinocytes, skin type V subject)

We agree that, according to published reports,^{8,9} free NADH can exhibit conformational heterogeneity, resulting in multi-exponential decays. While our instrument does not resolve these fine subcomponents, our solution measurements confirmed that the dominant lifetime of free NADH matches published values, supporting its use as a reference in our analysis. Additional control experiments with pure solutions can provide complementary validation, but they cannot replace tissue-based measurements, which are essential to capture fluorophore behavior in the native biological

environment. To better clarify our strategy and study rationale, we revised the opening of the Results section (p. 3), which now reads:

“Fluorescence lifetime detection was performed to analyze fluorescence lifetime properties of skin endogenous fluorophores across two spectral channels: A0 (405 - 506 nm) and A1 (506 - 610 nm). Two-photon excited fluorescence (TPEF) decays were transformed into phasor plots used to quantify fluorescence lifetimes and enhance molecular contrast by color-coding based on the measured phase lifetimes (Figure 1b). Reference phasor positions for the major skin fluorophores were established by combining published data with our own measurements (Figure 1c). Free NADH and free FAD were measured in solution with our system and validated against literature values,¹⁻⁴ while protein-bound NADH lifetime were adopted from published reports.²⁻⁴ In contrast, keratin and melanin lifetimes were determined *in vivo*, where specific physiological conditions allowed each to dominate the autofluorescence signal: keratin in the stratum corneum of a vitiligo subject (melanin absent) and melanin in basal keratinocytes of a highly pigmented skin type V subject. This strategy enabled us to obtain intrinsic phasor signatures of these fluorophores directly in human skin tissue, rather than relying solely on solution-based values.”

Question 2: Limited Sample Size and Lack of Statistical Validation:

The study includes only three healthy volunteers, one vitiligo case, and one seborrheic keratosis (SK) biopsy. No quantitative statistical analyses are provided for the presented data. I recommend either expanding the dataset to include more samples with appropriate statistical evaluation or rephrasing the conclusions to emphasize the exploratory nature of the study and outlining future directions in the introduction/discussion.

We agree with the reviewer’s comment and revised the conclusion as suggested. It now reads:

“In conclusion, this exploratory study demonstrates the potential of advanced imaging and phasor analysis to improve label-free molecular contrast in clinical skin imaging. While the dataset is limited, these findings provide a foundation for future larger-scale studies to assess the clinical utility of this approach.”

Question 3: The manuscript discusses cellular metabolic changes based on the unmixing of free and protein-bound NADH. However, tissue maintained *ex vivo* for up to 6 hours at room temperature is likely to undergo hypoxia and pH changes, all of which can shift metabolism toward glycolysis. These *ex vivo* artifacts make it challenging to interpret lifetime shifts as genuine metabolic changes. Supporting evidence from biochemical assays or literature clearly linking lifetime parameters to metabolic pathways and cellular viability should be included.

We agree that maintaining excised tissue *ex vivo* for several hours likely induces hypoxia and pH changes. Our intent in including this experiment was not to characterize physiological metabolism in viable tissue, but to provide a proof-of-concept demonstration that our 4-component unmixing approach is sensitive to metabolic shifts reflected in the NADH lifetime signal. The observed increase in free NADH is consistent with prior reports linking hypoxia and glycolytic states to shorter NADH lifetimes and a reduced protein-bound fraction.^{10,11} We therefore view these *ex vivo* changes not as a limitation, but as an opportunity to validate our methodology under conditions where metabolic perturbations are expected. While controlled *in vivo* metabolic studies would indeed be valuable, they are technically challenging; we used the *ex vivo* skin tissue experiment as a proof-of-concept, practical model for testing the sensitivity of our phasor-based analysis.

To better clarify this aspect, we revised the Results section (p. 14) to include the following paragraph:

“*Ex vivo* conditions inevitably induce hypoxia and pH changes. The lifetime shifts we observed in excised skin tissue are consistent with published findings that hypoxia shortens NADH lifetimes and decreases the protein-bound fraction in cells and tissue slices.^{10,11} Although not intended to replicate physiological metabolism, this experiment demonstrates the sensitivity of our approach to metabolic changes.”

Question 4: Missing Comparative Analysis:

Figure 5 lacks a comparison between phasor-FLIM and conventional diagnostic methods. Including histopathological analysis of the biopsy would strengthen the manuscript by providing a reference standard for comparison.

Figure 5 now includes histology comparison as shown below:

Figure 5: Applying Unmixing of Keratin and Melanin for the Detection of Seborrheic Keratosis (SK). a. Clinical image of the SK lesion. b. H&E-stained biopsy section showing a horn cyst (red arrow) surrounded by melanin-loaded keratinocytes (brown). Scale bar 25 μm . c. Intensity image acquired at the dermal-epidermal junction ($z=65 \mu\text{m}$) using FLAME in time-resolved detection mode. Inset highlights horn cysts. d. Corresponding lifetime phasor plots with gradient color distribution representing phase lifetime values. e. Phasor-colored image showing the distribution of phase lifetimes in the image. f. Combined color-coded maps from 4-component unmixing phasor analysis, showing keratin (red) and melanin (blue) distributions. This analysis distinctly identifies keratin-filled horn cysts (red arrows) from the surrounding melanin-rich keratinocytes (blue arrows) and highlights other morphological structures that exhibit varying melanin-to-keratin ratios (indicated by violet arrows). Scale bar is 300 μm .

Question 5: Incorrect Phasor Positions in Figure 1:

The positions of free and bound FAD are incorrect. Free FAD has a longer lifetime and should be located to the left side of the phasor plot, whereas protein-bound FAD should appear on the right. Additionally, the phasor position of SHG is not included in the figure and should be added.

We thank the reviewer for noting the error regarding the free and bound FAD positions on the phasor plot, which has now been corrected in Fig. 1c. Regarding SHG, it was not included in the figure because no SHG signal was detected in this study, which focused exclusively on fluorophores in the skin epidermis.

Minor Comments:

Minor Comment 1: State the reason for using a single excitation wavelength for all data acquisition.

We used a single-wavelength fiber laser in our compact, portable clinical multiphoton microscope to ensure suitability for clinical use. Compared to tunable lasers typically used in laboratory settings (e.g., Ti:Sapphire laser), a single-wavelength laser provides a smaller footprint, simpler operation, and greater robustness, making it more appropriate for clinical applications.

Minor Comment 2: Did you attempt to include FAD as a fifth fluorophore? If so, please explain why it was ultimately excluded.

In epidermal keratinocytes imaged in vivo in human skin, multiple fluorophores, including NAD(P)H and FAD (in their free and protein-bound forms), as well as keratin and melanin, contribute to the autofluorescence signal, complicating direct calculation of metabolic metrics. Incorporating both NADH and FAD alongside keratin and melanin would add substantial complexity to the unmixing process. To simplify the analysis, we selected NADH in addition to keratin and melanin. This choice was motivated by spectral considerations: FAD fluorescence strongly overlaps with keratin and melanin in the green channel, whereas the blue-shifted NADH fluorescence allowed us to design our detection channel A0 (405–506 nm) where FAD contribution is minimal. This strategy allowed us to work with NADH (free and protein-bound), keratin, and melanin in the 4-component unmixing analysis.

Minor Comment 3: In Figure 3, the interpretation that phasor contributions represent keratin is unclear. A "comet-like" structure indicating keratin is not consistently visible across all phasor plots, unlike melanin. Please clarify.

The term 'comet-like' describing the phasor distribution with a strong keratin contribution was borrowed from a previous publication reporting similar observations.⁷ This structure is most apparent in the phasor plots of the more superficial epidermal layers, where keratin dominates relative to other fluorophores. In contrast, melanin generally contributes more strongly than the other skin fluorophores, making its effect on the phasor distribution more pronounced in layers containing pigmented keratinocytes.

Minor Comment 4: In Figure 4.e, why do keratin and melanin distributions remain unchanged for skin type 3?

We revised Figure 4 to display the data as violin plots of the fractional intensity of keratin and melanin, which better show the distribution and depth-dependent trends.

In skin type III, the relative contributions of keratin and melanin appear balanced across the epidermis compared to skin type V (Figure 4d, previously Figure 4e). This likely reflects individual structural variations, such as keratin-rich skin folds and heterogeneous melanin localization rather than an effect specific to skin type.

Minor Comment 5: In Figure 6, there appears to be no change in NADH fractions across depths except at 45 μm for skin type 3. Could this be due to spectral bleed-through from keratin or melanin into the NADH signal?

A spectral bleed-through from keratin and melanin into the NADH channel is indeed present. Our unmixing strategy relies on time-resolved fluorescence and phasor analysis to separate the four key skin fluorophores (melanin, keratin, free and protein-bound NADH). In both skin types III and V, the shift in the protein-bound NADH fraction is relatively small, though slightly more pronounced in skin type V. The limited temporal resolution of our system captures the overall trend of this shift, but higher temporal resolution would allow more precise detection. To address the reviewer's comment, we revised Figure 6 to display the data as violin plots of the fractional intensity ratio (bound/total NADH), which better show the distribution and depth-dependent trends. These trends are now discussed in the text.

Minor Comment 6: Phasor plots across figures are densely packed and difficult to interpret. Please improve plot visibility for clarity.

We have replaced all the phasor plots in the paper for improved visibility.

Minor Comment 7: Briefly outline the diagnostic limitations of current label-free dermatologic imaging modalities (such as confocal microscopy, RCM, and OCT) and explain why molecular specificity is important. Additionally, consider including a comparison of the strengths and limitations (both technical and clinical) of FLAME relative to these conventional imaging techniques in the introduction or discussion section.

We revised the Introduction section to include the following paragraph:

“Current label-free dermatologic imaging modalities, such as reflectance confocal microscopy (RCM) and optical coherence tomography (OCT), provide high-resolution structural information but lack molecular specificity. RCM can be

combined with machine learning¹² to improve diagnostic performance, and fluorescence confocal approaches often require exogenous dyes.¹³ These limitations motivate the development of imaging techniques that combine high resolution, molecular contrast, and label-free detection, such as multiphoton microscopy (MPM).”

Reviewer #3 (Remarks to the Author):

A novel method to observe phasor characteristics to differentiate different fluorophores in MPM clinical skin imaging. This method uses FLIM as well as phasor analysis. ~4mm² at less than a minute. Compared to 100mm² in 1.25 minutes as this is one of the faster dermatology imaging methods

The imaging speed is far slower than other methods such as MS-DUV however these have a far lower specificity. These references could be mentioned:

J. Li, J. Garfinkel, X. Zhang, D. Wu, Y. Zhang, K. De Haan, H. Wang, T. Liu, B. Bai, Y. Rivenson, G. Rubinstein, P. O. Scumpia, A. Ozcan, Biopsy-free in vivo virtual histology of skin using deep learning. *Light Sci. Appl.* 10, 233 (2021).

J. Pérez-Anker, S. Ribero, O. Yélamos, A. Garcia-Herrera, L. Alos, B. Alejo, M. Combalia, D. Moreno- Ramirez, J. Malveyh, S. Puig, Basal cell carcinoma characterization using fusion ex vivo confocal microscopy: A promising change in conventional skin histopathology. *Br. J. Dermatol.* 182, 468–476 (2020).

We revised the Introduction section to include the following paragraph and suggested references:

“Current label-free dermatologic imaging modalities, such as reflectance confocal microscopy (RCM) and optical coherence tomography (OCT), provide high-resolution structural information but lack molecular specificity. RCM can be combined with machine learning¹² to improve diagnostic performance, and fluorescence confocal approaches often require exogenous dyes.¹³ These limitations motivate the development of imaging techniques that combine high resolution, molecular contrast, and label-free detection, such as multiphoton microscopy (MPM).”

Question 1: Figure 1: Good informative figure. Each subfigure proceeds to add more details. E) was a little more difficult to understand at first glance. Embellishing the highlighted cells would help convey the message.

Figure 1e was designed to show the relative contributions of the four key fluorophores across the epidermis. To avoid overcrowding an already information-rich figure, we have not added additional embellishments. Instead, we clarified in the figure legend how the highlighted cells correspond to specific fluorophore contributions, to help guide the reader in interpreting the figure. The Figure 1e legend now reads:

“Figure 1. Multiphoton Microscopy of Skin Layers and Phasor-Based FLIM Analysis. (...) e, Schematic depiction of phasor analysis highlighting the unmixing of four distinct tissue components based on their phasor signatures. Fractional intensity maps are generated for each component, visualizing their spatial distribution. **Highlighted regions indicate representative areas where each component predominates, aiding interpretation of the unmixing results.”**

Question 2: Line 106: The “four distinct components to the total phasor distribution “. It would be beneficial for the reader to explicitly explain what these components are or might be.

We revised the paragraph including line 106 to explicitly identify the four components. It now reads:

“...we focused on separating four distinct components: free and protein-bound NADH from keratin and melanin (Figure 1e).”

Question 3: Figure 3: d depth info should be larger font size. Phasor plots are very hard to read the values. A value to represent the change observed in each of the different depths would be helpful to approximate the observed lifetime.

We appreciate the reviewer’s suggestion. Figures 2 and 3 have been modified for legibility, including increasing images and font sizes, and adding higher resolution phasor pictures. We have also modified the phasor plots with contour graphs for proper display where it helped us to find component positions.

We appreciate the reviewer’s comment about the observed lifetime. However, the phasor representation does not allow for the calculation of a simple average value. Given that the distributions are often broad and irregular—unlike the more compact, elliptical shapes typically observed in cellular measurements—determining a central position is neither straightforward nor particularly informative in this context.

Question 4: Figure 5: It seems like the intensities and Keratin-Melanin map are very similar. It would be good to show how these plots similar/different using some sort of structural imaging metric such as the SSIM, a histogram, Pearson’s correlation coefficient, or Manders split coefficients.

The observed similarity is expected. In our 4-component unmixing approach, each pixel is decomposed based on the relative contributions of keratin, melanin, free NADH, and protein-bound NADH. Since keratin and melanin are more abundant than the NADH species in these tissue samples, most pixels naturally exhibit high fractional contributions from these dominant components, creating the expected visual similarity with total intensity.

To illustrate subtle but meaningful differences, we revised Figure 5 to include insets highlighting areas where the keratin-melanin map diverges from the intensity distribution. We appreciate the recommendation to include structural imaging metrics. However, we believe the visual comparison in the revised figure sufficiently illustrates the key differences for this study's scope.

Question 5: Line 298: The differentiation of the graphs in figure 6, d, e, f should be quantified with numerical values.

The shift in the protein-bound NADH fraction measured across the epidermal layers is relatively small, though slightly more pronounced in skin type V. The limited temporal resolution of our system captures the overall trend of this shift, but higher temporal resolution would allow more precise detection. To address the reviewer’s comment, we revised Figure 6 to display the data as violin plots of the fractional intensity ratio (bound/total NADH) with median values, which better show the distribution and depth-dependent trends. These trends are now discussed in the text.

Question 6: Line 377: The strategy should be shown here or referenced.

In the Discussion section including line 377, we revised the text to include references related to the strategy used and to clearly identify the components: “By employing a four-component unmixing strategy,¹⁴⁻¹⁸ targeting free and protein-bound NADH, keratin, and melanin, we quantified the spatial distributions of keratin and melanin across various epidermal depths in subjects representative of different skin types.”

Question 7: Line 383: Quantification in regards to the accuracy of this unmixing process should be characterized against a conventional imaging technology. The tissue biopsy could be sectioned into two adjacent sections. One section could be stained with specific molecular specific fluorophores that are identified, such as keratin, NADH, and melanin. This would allow metrics such as the accuracy and specificity of this new imaging approach to be quantified.

We thank the reviewer for this suggestion. While direct validation against molecularly stained adjacent tissue sections would provide an additional metric of accuracy, the primary goal of this study was to demonstrate the feasibility and sensitivity of the 4-component phasor unmixing approach for label-free human skin imaging. The validity of our approach is supported by biologically consistent results:

- Fractional intensity maps reflect biologically consistent distributions, such as keratin enrichment in the stratum corneum and skin folds, and melanin localization in basal layers (Figure 4).
- In a clinically relevant case, keratin–melanin maps correctly highlighted features such as keratin pearls and pigmented cells, consistent with histology (Figure 5).
- Ex vivo experiments showed shifts in free and bound NADH lifetimes that align with published findings (Figure 7).

Prior studies (e.g., Ref.¹⁹) have also validated the origin of label-free FLIM signals by direct comparison with histology. Direct biopsy-stain comparisons, as suggested, are technically challenging due to the mismatch between 3D tissue volumes and 2D sections. Instead, our group has focused on in vivo validation in clinically relevant contexts. For example, in Ref.²⁰, we showed that FLAME imaging of human skin can differentiate Fitzpatrick skin types using time-resolved detection (though without phasor unmixing). A similar study could be extended to validate phasor-based quantification of melanin, but such work is beyond the scope of this proof-of-concept study.

Question 8: Line 387: A broader comparison could be discussed here with comparisons against MS- DUV, RCM, and other NLM methods.

We expanded the Introduction section to include a comparison with RCM and OCT as they are the most widely used and clinically established label-free imaging modalities in dermatology. As noted in the revised text, both techniques provide valuable high-resolution structural information but lack intrinsic molecular specificity. We now explicitly include this broader context in the Introduction section of the manuscript:

“Current label-free dermatologic imaging modalities, such as reflectance confocal microscopy (RCM) and optical coherence tomography (OCT), provide high-resolution structural information but lack molecular specificity. RCM can be combined with machine learning¹² to improve diagnostic performance, and fluorescence confocal approaches often require exogenous dyes.¹³ These limitations motivate the development of imaging techniques that combine high resolution, molecular contrast, and label-free detection, such as multiphoton microscopy (MPM).”

Question 9: Overall, this paper reads well with some minor clarifications needed, as well as the addition of comparisons to other imaging methods. The paper talks about verifying the composition of their imaging technique, but this is only done by assuming certain areas imaged having an expected molecular composition. A more in-depth verification method would provide the reader with greater confidence in the abilities of this new imaging Method.

We thank the reviewer for this comment. We agree that more direct validation methods would further strengthen confidence in our approach. However, the goal of this study was to establish a proof-of-concept for 4-component phasor unmixing in label-free human skin imaging. The validity of our results is supported by biologically consistent spatial patterns, histological correspondence in clinical samples, and agreement with previously reported results. These findings demonstrate the biological relevance of our approach and provide a foundation for future work incorporating additional validation strategies.

References:

- 1 Lakowicz, J. R., Szmajdzinski, H., Nowaczyk, K. & Johnson, M. L. Fluorescence lifetime imaging of free and protein-bound NADH. *Proc Natl Acad Sci U S A* **89**, 1271-1275, doi:10.1073/pnas.89.4.1271 (1992).
- 2 Datta, R., Heaster, T. M., Sharick, J. T., Gillette, A. A. & Skala, M. C. Fluorescence lifetime imaging microscopy: fundamentals and advances in instrumentation, analysis, and applications. *J Biomed Opt* **25**, 1-43, doi:10.1117/1.JBO.25.7.071203 (2020).
- 3 Datta, R., Alfonso-García, A., Cinco, R. & Gratton, E. Fluorescence lifetime imaging of endogenous biomarker of oxidative stress. *Sci Rep* **5**, 9848, doi:10.1038/srep09848 (2015).

- 4 Ranjit, S., Malacrida, L., Stakic, M. & Gratton, E. Determination of the metabolic index using the fluorescence lifetime of free and bound nicotinamide adenine dinucleotide using the phasor approach. *J Biophotonics* **12**, e201900156, doi:10.1002/jbio.201900156 (2019).
- 5 Palero, J. A., de Bruijn, H. S., van der Ploeg van den Heuvel, A., Sterenborg, H. J. & Gerritsen, H. C. Spectrally resolved multiphoton imaging of in vivo and excised mouse skin tissues. *Biophys J* **93**, 992-1007, doi:10.1529/biophysj.106.099457 (2007).
- 6 Muir, R., Forbes, S., Birch, D. J. S., Vyshemirsky, V. & Rolinski, O. J. Keratin intrinsic fluorescence as a mechanism for non-invasive monitoring of its glycation. *Methods and Applications in Fluorescence* **11**, 015003, doi:10.1088/2050-6120/aca507 (2023).
- 7 Pena, A. M. *et al.* In vivo melanin 3D quantification and z-epidermal distribution by multiphoton FLIM, phasor and Pseudo-FLIM analyses. *Sci Rep* **12**, 1642, doi:10.1038/s41598-021-03114-0 (2022).
- 8 Visser, A. J. W. G. & Hoek, A. v. THE FLUORESCENCE DECAY OF REDUCED NICOTINAMIDES IN AQUEOUS SOLUTION AFTER EXCITATION WITH A UV-MODE LOCKED Ar ION LASER. *Photochemistry and Photobiology* **33**, 35-40, doi:<https://doi.org/10.1111/j.1751-1097.1981.tb04293.x> (1981).
- 9 Li, H. *et al.* Ultrafast fluorescence dynamics of NADH in aprotic solvents: Quasi-static self-quenching unmasked. *Journal of Photochemistry and Photobiology A: Chemistry* **436**, 114384, doi:<https://doi.org/10.1016/j.jphotochem.2022.114384> (2023).
- 10 Vishwasrao, H. D., Heikal, A. A., Kasischke, K. A. & Webb, W. W. Conformational dependence of intracellular NADH on metabolic state revealed by associated fluorescence anisotropy. *J Biol Chem* **280**, 25119-25126, doi:10.1074/jbc.M502475200 (2005).
- 11 Zbinden, A. *et al.* Fluorescence lifetime metabolic mapping of hypoxia-induced damage in pancreatic pseudo-islets. *J Biophotonics* **13**, e202000375, doi:10.1002/jbio.202000375 (2020).
- 12 Li, J. *et al.* Biopsy-free in vivo virtual histology of skin using deep learning. *Light: Science & Applications* **10**, 233, doi:10.1038/s41377-021-00674-8 (2021).
- 13 Pérez-Anker, J. *et al.* Basal cell carcinoma characterization using fusion ex vivo confocal microscopy: a promising change in conventional skin histopathology. *Br J Dermatol* **182**, 468-476, doi:10.1111/bjd.18239 (2020).
- 14 Vallmitjana, A., Torrado, B., Dvornikov, A., Ranjit, S. & Gratton, E. Blind Resolution of Lifetime Components in Individual Pixels of Fluorescence Lifetime Images Using the Phasor Approach. *The Journal of Physical Chemistry B* **124**, 10126-10137, doi:10.1021/acs.jpcc.0c06946 (2020).
- 15 Vallmitjana, A., Lepanto, P., Irigoien, F. & Malacrida, L. S. Multi-harmonic spectral phasor analysis of LAURDAN hyperspectral imaging enables the in vivo study of membrane dynamics of multiple membrane-bound organelles. *Biophysical Journal* **122**, 153a, doi:10.1016/j.bpj.2022.11.990 (2023).
- 16 Vallmitjana, A., Lepanto, P., Irigoien, F. & Malacrida, L. Phasor-based multi-harmonic unmixing for in-vivo hyperspectral imaging. *Methods and Applications in Fluorescence* **11**, 014001, doi:10.1088/2050-6120/ac9ae9 (2023).
- 17 Vallmitjana, A. *et al.* Resolution of 4 components in the same pixel in FLIM images using the phasor approach. *Methods and Applications in Fluorescence* **8**, 035001, doi:10.1088/2050-6120/ab8570 (2020).
- 18 Ranjit, S., Datta, R., Dvornikov, A. & Gratton, E. Multicomponent Analysis of Phasor Plot in a Single Pixel to Calculate Changes of Metabolic Trajectory in Biological Systems. *The Journal of Physical Chemistry A* **123**, 9865-9873, doi:10.1021/acs.jpca.9b07880 (2019).
- 19 Fast, A. *et al.* Fast, large area multiphoton exoscope (FLAME) for macroscopic imaging with microscopic resolution of human skin. *Sci Rep* **10**, 18093, doi:10.1038/s41598-020-75172-9 (2020).
- 20 Vicente, J. R., Durkin, A., Shrestha, K. & Balu, M. In vivo imaging with a fast large-area multiphoton exoscope (FLAME) captures the melanin distribution heterogeneity in human skin. *Sci Rep* **12**, 8106, doi:10.1038/s41598-022-12317-y (2022).

Response to reviewers – Final Submission

We appreciate the comments from the reviewers and addressed the final corrections as follows:

Reviewer 2

Q1. *Reviewer 2 raised a question about the phasor coordinates used in the multicomponent analysis.*

A1. We have added the following table to the supplemental material that shows the phasor coordinates of the pure component positions for the calculation.

Species	Lifetime (ns)	G	S
Free NADH	0.4 ns	0.96	0.20
Protein-bound NADH	3.4 ns	0.25	0.44
Keratin	1.1 ns	0.77	0.42
Melanin	0.0 ns	1.00	0.00

Supplemental Table 1. G,S coordinates from the measured and published data. The G,S coordinates, calculated from measurement *in vitro* (free and protein bound NADH) and *in vivo* (Keratin and Melanin), were used to calculate the component distribution.

Reviewer 3

Q4. *I believe that the addition of the contrasting maps below the original figures and violin plots give improved context to this image. It is interesting to see the distribution change over the different depths. One improvement is to increase the font size for the phasor plot. Another important distinction to make is to explicitly say if the figure intensities are normalized in each individual image or if they are relative to each other.*

A4. We thank the reviewer for the positive feedback on the contrasting maps and violin plots. Regarding the phasor plot font size, we have increased it to the maximum size allowed by the available space while maintaining figure clarity. We note that the primary information conveyed by the phasor plot is the phase lifetime values, which are color-coded for easier interpretation. To address the reviewer's second point, we have now explicitly stated in the Methods section that all figure intensities are normalized individually for each image, rather than being displayed relative to one another

Q7. *I agree that this info may go above and beyond the scope of this manuscript. Figure 5 with the keratin-melanin maps helps to address my concerns. I believe that this would be a valuable talking point in the discussion and would encourage the writers to add discussion points including: prior study validation of FLIM, explaining the proof of concept by referring the structural images in figure 5 (I see this is present but adding further details highlighting why this is important and references demonstrating these structural components would be good to include).*

A7. We have expanded the Discussion to include additional context highlighting both prior validation of FLIM approaches for melanin quantification and the novelty of our current work. Specifically, we now clarify that, while previous studies have demonstrated *in vivo* FLIM and phasor analysis for assessing melanin content in skin, to our knowledge, no prior work has demonstrated unmixing of multiple

endogenous fluorophores, including melanin, keratin, and metabolic cofactors (NADH, FAD), to generate molecular maps in human skin.

We have revised the Discussion paragraph (page 16) to read as follows:

“While previous studies have described approaches for melanin quantification in human skin using in vivo FLIM imaging [Ref 14] and phasor analysis [Ref 27], the present study represents, to our knowledge, the first demonstration of unmixing endogenous skin components such as melanin and keratin, enabling visualization of distinct biochemical and structural features within a lesion. To illustrate the clinical applicability of our unmixing technique, we applied it to a freshly excised biopsy subsequently diagnosed as macular seborrheic keratosis (SK). The fractional intensity distribution maps generated through phasor analysis distinguished critical features, such as keratin-filled "horn cysts" from other structures exhibiting varying melanin-to-keratin ratios. The ability to identify these key structural features label-free demonstrates the potential of this unmixing approach for distinguishing benign pigmented lesions from melanoma and for advancing real-time optical biopsy applications.”

Q8. *I appreciate the added comparison technology context in the introduction of the manuscript. However, the request for a broader comparative discussion remains insufficiently addressed. I recommend adding a concise comparison in the Discussion to contextualize the technique’s relative strengths and limitations.*

A8. We thank the reviewer for this valuable suggestion. In response, we have expanded the Discussion section to include a concise comparative paragraph that contextualizes the relative strengths and limitations of MPM compared to other label-free imaging modalities used for in vivo human skin imaging.

The newly added text (Discussion section, page 17) reads as follows:

Compared to other label-free dermatologic imaging modalities, such as RCM, OCT, line field confocal OCT (LC-OCT), MPM offers distinct advantages in combining submicron resolution with intrinsic molecular contrast derived from endogenous fluorophores. RCM enables rapid, wide-field en-face imaging with sub-cellular resolution and is widely used in clinical screening, but its contrast originates primarily from refractive index variations, limiting molecular specificity. [Ref 45- **PMID: 27785781**] OCT provides deeper penetration (up to 1–2 mm) and quantitative structural information, yet it lacks cellular-level resolution. [Ref 46- **PMID: 29701018**] LC-OCT enables rapid, wide-field cross-sectional imaging and sub-cellular resolution, but the absence of molecular specificity limits its use primarily to morphological assessments. [Ref 47- **PMID: 38137869**] In contrast, MPM provides optical sectioning and molecular-level information through TPEF and SHG signals, allowing the visualization of biochemical and architectural features with histology-like, submicron detail. Current limitations of MPM include slower imaging speed and higher system complexity compared to RCM and OCT; however, ongoing developments in fast scanning approaches, compact femtosecond lasers, and phasor-based analysis are rapidly improving its translational potential. Collectively, these complementary characteristics position MPM as a molecularly informative extension to existing imaging modalities, offering unique opportunities for real-time, non-invasive optical biopsy and longitudinal monitoring of skin disease. [Ref 48-**PMID: 41112782**]